# CAT-Q: Cost-efficient and Accurate Ternary Quantization for LLMs

Shigeng Wang [* 1]  Chao Li [* 1]  Yangyuxuan Kang [1]  Jiawei Fan [1]  Anbang Yao [1]

## Abstract

In this paper, we present CAT-Q, **C**ost-efficient and **A**ccurate **T**ernary **Q**uantization, for compressing and accelerating LLMs. Unlike existing state-of-the-art ternary quantization methods that rely on data-intensive and costly quantization-aware training to mitigate severe performance degradation, CAT-Q is a simple yet effective post-training quantization scheme that is readily applicable to LLMs with diverse architectures and model sizes. It has two key components, learnable modulation (LM) and softened ternarization (ST), which are coupled from an optimization perspective. LM leverages a composition of learnable factors to modulate the distribution of pre-trained high-precision weights and the ternary threshold, making them less sensitive to ternarization. ST further introduces a differentiable transition function to guide the ternarization process toward stable convergence. We show that, for pre-trained LLMs with 1.7B to 8B parameters, CAT-Q can efficiently quantize them into ternary models using only 512 calibration samples, while achieving superior performance than the seminal BitNet 1.58-bit v1 and v2 families (with 1.3B to 7B parameters) trained with 100B tokens, yielding about a 100,000× reduction in training tokens. Moreover, we show for the first time that CAT-Q can quantize much larger pre-trained LLMs having 14B to 235B parameters into leading ternary models within just 8 to 60 hours on 8 A100-80GB GPUs. Code is available at https://github.com/IntelChina-AI/BitTern.

## 1. Introduction

Large language models (LLMs) (Vaswani et al., 2017; Brown et al., 2020; Achiam et al., 2023; Anil et al., 2023; Liu et al., 2024a; Hurst et al., 2024; Jaech et al., 2024; Yang et al., 2024; Guo et al., 2025; Yang et al., 2025; Singh et al., 2025; Comanici et al., 2025) have demonstrated remarkable performance across a wide range of language modeling and reasoning tasks. However, their large sizes incur significant memory and computational costs, posing a major obstacle to deploy them in real-world applications, especially on resource-constrained devices. Various techniques (Zafrir et al., 2019; Ma et al., 2023; Sun et al., 2024; Sanh et al., 2019; Hsieh et al., 2023; Zhang et al., 2021; Hsu et al., 2022) have been proposed to reduce model size and accelerate inference, among which quantization is particularly appealing due to its effectiveness and ease of implementation.

The ultimate optimization objective of quantization is to reduce the numerical precision of model parameters while preserving model performance, either via post-training quantization (PTQ) or quantization-aware training (QAT). Most existing methods (Shen et al., 2020; Dettmers et al., 2022; Yao et al., 2022; Frantar et al., 2023; Xiao et al., 2023; Shao et al., 2024; Lin et al., 2024b; Chee et al., 2024; Ashkboos et al., 2024b; Sun et al., 2025; Wang et al., 2026) rely on PTQ, which quantizes pre-trained LLMs using a small set of calibration samples and does not require the highly expensive training or re-training pipeline used in QAT, and are therefore easy to use in practice. These PTQ methods are generally effective in 8-bit quantization, and some, such as QuaRot (Ashkboos et al., 2024b), FlatQuant (Sun et al., 2025) and SliderQuant (Wang et al., 2026), attain favorable performance even at 4-bit precision. However, they still exhibit poor performance in ultra-low-bit settings, such as 2-bit quantization.

In this paper, we focus on ternary (also known as 1.58-bit) quantization, an extreme form of quantization, which maps high-precision weights to $\{1, 0, -1\}$. It brings more than a $10\times$ reduction in memory consumption relative to its FP16 counterpart. Moreover, by introducing sparsity through the zero state, ternary quantization retains the hardware-friendly arithmetic advantage of binary quantization (Courbariaux et al., 2015; Rastegari et al., 2016; Xu et al., 2024; Huang et al., 2024; Shang et al., 2024; Li et al., 2025), whereby most expensive floating-point multiplication operations can be replaced by far more energy-efficient integer additions and subtractions, while offering richer expressive capacity. For ternary quantization, it is challenging to devise an effec-

---

[*]Equal contribution  [1]Intel Labs China. This work was done when Shigeng Wang was an intern at Intel Labs China. Anbang Yao conceived the project and led the writing of the paper. Correspondence to: Anbang Yao <anbang.yao@intel.com>.

*Proceedings of the 43 rd International Conference on Machine Learning*, Seoul, South Korea. PMLR 306, 2026. Copyright 2026 by the author(s).

tive optimization scheme that can accurately approximate high-precision weights by only using values of $\{1, 0, -1\}$ due to the high risk of information loss. To alleviate severe accuracy drop in a more straightforward manner, existing state-of-the-art ternary quantization methods, such as BitNet 1.58-bit (Ma et al., 2024; Wang et al., 2025), TriLM (Kaushal et al., 2025) and Tequila (Huang et al., 2026), unanimously adopt QAT, which enables the model to adapt its parameters to 1.58-bit precision by simulating ternarization during full-precision training. Despite achieving promising results, they require massive amounts of training tokens (e.g., hundreds of billions) and entail enormous computational, energy and time costs, limiting their scalability and practicality. For example, the BitNet 1.58-bit v2 family (Wang et al., 2025) is trained on 100B tokens, with its largest model containing 7B parameters; while the TriLM family (Kaushal et al., 2025) is trained on 300B tokens, with a maximum model size of 3.9B parameters. Tequila (Huang et al., 2026), the most recent ternary quantization method, still requires 10B tokens to quantize a pre-trained LLM into a ternary model via re-training. Furthermore, Tequila is merely evaluated on two small-scale models: the 1B and 3B instances of the Llama3 family (Grattafiori et al., 2024). In a nutshell, existing top-performing ternary quantization methods are tailored to a narrow set of LLMs with specific architectures and small model scales, owing to their data-intensive and costly QAT pipelines. Although PTQ is preferred in industry as it does not require access to the original training data and eliminates data privacy and security concerns, as well as the need for massive computational resources, there currently exists no PTQ-based ternarization method that could match the performance of the aforementioned QAT-based methods. To bridge this gap, we revisit PTQ for ternary weight quantization from an optimization perspective of studying its major technical challenges.

Given a small calibration set (e.g., 512 sequences of length 2048 tokens randomly sampled following common practice in the PTQ field, rather than using the original training data) and a pre-trained LLM with arbitrary architecture and model scale, we identify that two major challenges are closely related to the severe performance degradation observed when minimizing reconstruction error of ternary weight quantization. The first challenge is that direct ternarization of pre-trained high-precision weights tends to generate a ternary weight distribution that is not well aligned with the high-precision weight distribution. This distributional misalignment between the resulting ternary weights and their high-precision counterparts leads to serious information loss. Therefore, designing a proper strategy to refine the distributional alignment is critical to reduce reconstruction error. The second challenge, which is more fundamental, is that ternarization optimization is notably difficult to converge, due to its extremely low-bit form and non-differentiable

nature. To address this extreme discretization problem, existing ternary quantization methods for LLMs follow previous strategies (Li et al., 2016; Zhu et al., 2017) originally designed for ternarizing convolutional neural networks containing significantly fewer parameters. Specifically, they always use non-differentiable hard ternarization functions throughout the optimization process, which cannot adapt to complex weight variations and ensure stable convergence, especially in the PTQ regime.

Motivated by the above analysis, we present **C**ost-efficient and **A**ccurate **T**ernary **Q**uantization (CAT-Q), a new post-training ternarization method that is applicable to a wide range of pre-trained LLMs. CAT-Q consists of two key components. The first component, learnable modulation (LM), leverages a composition of three learnable factors to refine the distributional statistics of pre-trained high-precision weights as well as the ternary threshold based on a small set of calibration samples. These learnable factors enable an adaptive weight distribution transformation that suppresses the effect of outliers and makes high-precision weights less sensitive to ternarization, mitigating the distributional misalignment between the resulting ternary weights and their high-precision counterparts, to a certain extent. The second component, softened ternarization (ST), introduces a novel transition function that guides the ternarization process toward stable convergence via establishing a two-stage relay of differentiable ternarization and hard ternarization. Coupling LM and ST into a sliding-layer quantization pipeline results in a clean and efficient implementation of CAT-Q.

We conduct comprehensive experiments to evaluate the efficacy of CAT-Q across 10 pre-trained LLMs spanning diverse architectures (including dense and mixture of experts (MoE) models) and model scales (1.7B to 235B parameters) (Yang et al., 2024; 2025; Touvron et al., 2023). We show that, under similar model scales (1.7B to 8B parameters), ternary weight models generated by CAT-Q using only 512 calibration samples (about 1 million tokens) outperform the seminal BitNet 1.58-bit v1 and v2 families trained with 100B tokens, bringing an approximately 100,000× reduction in training tokens. Moreover, we show for the first time that CAT-Q can readily scale to much larger pre-trained LLMs having 14B to 235B parameters. When quantizing LLMs with 1.7B to 235B parameters, CAT-Q requires only 1 to 60 hours on 8 A100-80GB GPUs. In addition, our models with ternary weights and 8-bit activations get performance comparable to their ternary-weight-only counterparts, further demonstrating the broader applicability of our method.

## 2. Method

Figure 1 shows an overview of CAT-Q. Next, we describe its formulation and key components.

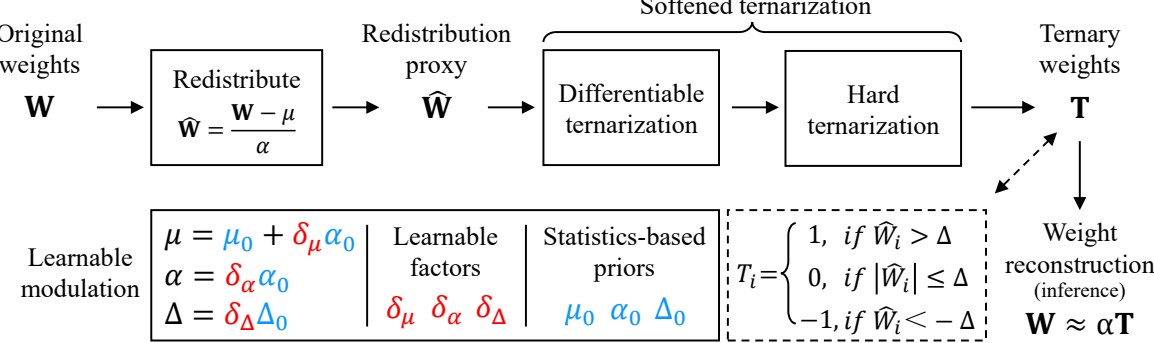

*Figure 1.* Overview of the CAT-Q's learning flow for ternarizing the weights of a linear layer in any pre-trained LLM and its hardware-friendly weight reconstruction for ternary model deployment. Please see Figure 3 for an illustration of the softened ternarization process.

### 2.1. Preliminary Concepts and Motivation

The concept of ternary weight quantization was originally proposed in TWN (Li et al., 2016) to train convolutional neural networks from scratch for computer vision tasks. Concretely, it constrains model weights to $\{1, 0, -1\}$ via solving a layer-wise weight reconstruction problem:

$$\underset{\alpha, \mathbf{T}}{argmin} \, ||\mathbf{W} - \alpha\mathbf{T}||_2^2. \tag{1}$$

Here $\mathbf{W}$ denotes the high-precision weights for a linear layer, and $\alpha > 0$ denotes a scaling factor to rescale the corresponding ternary weights $\mathbf{T}$ whose elements are obtained by a hard ternarization function:

$$T_i = Q(W_i; \Delta) = \begin{cases} 1, & if \ W_i > \Delta \\ 0, & if \ |W_i| \leq \Delta \\ -1, & if \ W_i < -\Delta, \end{cases} \tag{2}$$

where $W_i$ denotes the $i^{th}$ element of $\mathbf{W}$ and $\Delta > 0$ is a threshold. Subsequent works for ternary LLMs typically follow TWN. TernaryBERT (Zhang et al., 2020) makes an early research effort to ternarize the weights of small BERT-based language models (Devlin et al., 2019) during fine-tuning, which uses transformer distillation (Jiao et al., 2020) to compensate for severe accuracy degradation. Instead, existing state-of-the-art ternary LLMs, such as BitNet 1.58-bit (Ma et al., 2024; Wang et al., 2025), TriLM (Kaushal et al., 2025) and Tequila (Huang et al., 2026), rely on QAT. In contrast to these works, we focus on ternarizing the weights of LLMs in the more challenging PTQ regime, aiming to strike a substantially better balance between quantization cost and performance, and enabling its broad applicability to LLMs with diverse architectures and model sizes.

### 2.2. Learnable Modulation

According to the above formulation, the core problem in ternary weight quantization is how to estimate appropriate values for the scaling factor $\alpha$ and the threshold $\Delta$. The pioneering TWN approximates them as $\alpha = \frac{1}{|I_\Delta|} \sum_{i \in I_\Delta} |W_i|$

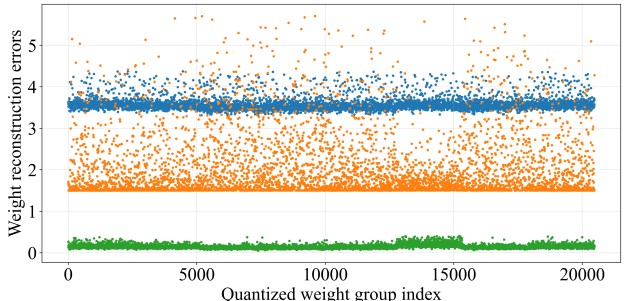

*Figure 2.* Comparison of weight reconstruction errors with the scaling factor $\alpha$ and the threshold $\Delta$ determined by static approximation (blue dots), direct learning (orange dots) and our learnable modulation (green dots). Under the same settings, we use the $4^{th}$ layer of Qwen3-4B for an illustration. *In the Appendix, we provide additional comparisons on different layers across multiple LLMs.*

and $\Delta = \frac{0.7}{n} \sum_{i=1}^{n} |W_i|$, where $I_\Delta = \{i \mid 1 \leq i \leq n, |W_i| > \Delta\}$, $|I_\Delta|$ is the number of elements in $I_\Delta$, and $n$ is the number of elements in $\mathbf{W}$. Recent ternary LLMs, such as BitNet 1.58-bit (Ma et al., 2024; Wang et al., 2025), TriLM (Kaushal et al., 2025) and Tequila (Huang et al., 2026), simply use the absmean ternarization: $\alpha = \frac{1}{n} \sum_{i=1}^{n} |W_i|$ and $\Delta = \frac{\alpha}{2}$. We notice that some prior works (Rastegari et al., 2016; Zhu et al., 2017; Chen et al., 2024b) directly treat the scaling factor $\alpha$ and or the threshold $\Delta$ as learnable parameters to train binary or ternary neural networks, while others (Esser et al., 2020; Gong et al., 2019; Liu et al., 2025b) extend such learning paradigm to estimate quantization step sizes under different bit-width settings. Inspired by them, we also learn $\alpha$ and $\Delta$ for ternarizing each weight group of pre-trained LLMs based on a small number of calibration samples. However, in the PTQ regime, we empirically find that directly learning $\alpha$ and $\Delta$ still suffers from the distributional misalignment between the resulting ternary weights and their pre-trained high-precision counterparts, showing only modest improvement over static approximation to alleviate severe accuracy degradation (see Figure 2 for an illustrative comparison). To address this

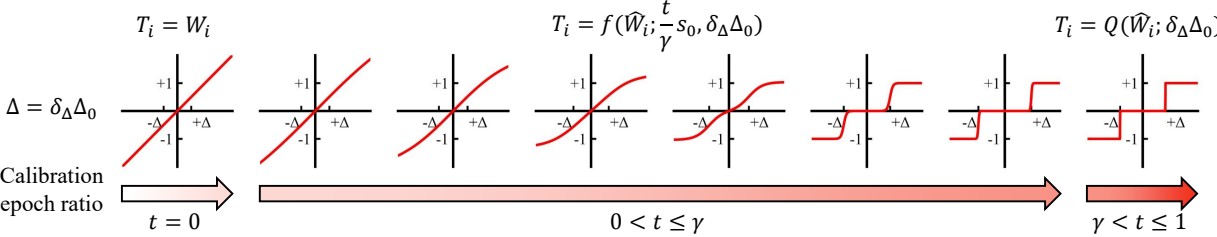

*Figure 3.* Illustration of the softened ternarization (ST) process. For a linear layer, taking its pre-trained weights $\mathbf{W}$ as the initialization point ($t = 0$), ST employs a learnable two-stage relay of differentiable ternarization and hard ternarization to ensure stable convergence. In the first stage, ST produces an asymptotic ternary output by performing continuous quantization based on the transformed weights $\hat{\mathbf{W}}$. It relies on a novel smooth transition function $f(\cdot)$ to gradually evolve from the identity mapping to differentiable ternarization via a sequence of continuous mappings with progressively increasing sharpness along the normalized calibration time state $0 < t \leq \gamma$. In the second stage $\gamma < t \leq 1$, ST proceeds with hard ternarization to get a final solution. Notations are clarified in the Method section.

issue, we present learnable modulation (LM), the first module of our method, which modulates the pre-trained weight distribution to be less sensitive to the ternarization via introducing a learnable linear transformation defined as:

$$\hat{\mathbf{W}} = \frac{\mathbf{W} - \mu}{\alpha}, \text{ where } \mu = \mu_0 + \delta_\mu \alpha_0, \ \alpha = \delta_\alpha \alpha_0. \quad (3)$$

Here, $\mu_0 = \frac{1}{n} \sum_{i=1}^n W_i$ denotes the mean of $\mathbf{W}$, $\alpha_0 = \frac{1}{n} \sum_{i=1}^n |W_i - \mu_0|$ is the absolute mean of $\mathbf{W} - \mu_0$, and $-1 < \delta_\mu < 1$ and $\delta_\alpha > 0$ are two learnable factors to refine $\alpha_0$ and $\mu_0$, which also enable the elements of the transformed weights $\hat{\mathbf{W}}$ can have reversed signs against the original weights $\mathbf{W}$. Besides, for the ternarization function, we have $\Delta = \delta_\Delta \Delta_0$, where $\delta_\Delta > 0$ is a learnable factor to adjust the initial threshold $\Delta_0$. By default, $\Delta_0 = 0.5$ as we use the transformed weights $\hat{\mathbf{W}}$ (equivalently, $\Delta_0 = \frac{\alpha}{2}$ corresponding to the pre-trained high-precision weights $\mathbf{W}$) to determine $\mathbf{T}$. We introduce a disentangled learning strategy to determine these three factors $\delta_\mu$, $\delta_\alpha$ and $\delta_\Delta$. Specifically, we use the transformed weights $\hat{\mathbf{W}}$ as a redistribution proxy to learn the threshold $\Delta$ and the ternary weights $\mathbf{T}$, while approximating the pre-trained weights as $\mathbf{W} \approx \alpha \mathbf{T}$ without $\mu$. This disentangled learning strategy not only preserves the original hardware-friendly implementation property of TWN but also yields better ternarization performance than the counterpart that approximates $\mathbf{W} \approx \alpha \mathbf{T} + \mu$, as validated by an ablation shown in Table 9. Mathematically, following the form of the ternarization function $Q(\cdot)$ defined in Equation 2, in the LM component, the $i^{th}$ element of the ternary weights $\mathbf{T}$ is obtained by:

$$T_i = Q(\hat{W}_i; \delta_\Delta \Delta_0). \quad (4)$$

## 2.3. Softened Ternarization

Due to its extreme low-bit form and non-differentiable nature, ternary weight quantization is inherently difficult to optimize and converge. Following TWN (Li et al., 2016), existing QAT-based ternarization methods for LLMs (Ma

et al., 2024; Wang et al., 2025; Kaushal et al., 2025; Huang et al., 2026) perform hard ternarization using the straight-through estimator for gradient approximation, where convergence is largely facilitated by the quantization-aware training process, particularly for LLMs with small model scales. However, in the post-training quantization (PTQ) regime, where no constraints are imposed on model architectures or scales, this fundamental challenge becomes significantly more pronounced. Although, in Figure 2, we have shown that learning the scaling factor $\alpha$ and the threshold $\Delta$ with the LM component achieves better performance than static approximation and direct learning methods, it still performs hard ternarization (facing the non-differentiability issue) throughout the calibration learning, lacking a technical guarantee for good convergence. To bridge this gap, we propose softened ternarization (ST), the second component of our method inspired by previous works for quantizing convolutional neural networks (Gong et al., 2019; Esser et al., 2020), which relies on a novel transition function:

$$f(W; s, \Delta) = \frac{\tanh(s(W - \Delta)) + \tanh(s(W + \Delta))}{2 \tanh(s)}. \quad (5)$$

Here, $W$ denotes an arbitrary weight input either from the pre-trained weights $\mathbf{W}$ or its transformed version $\hat{\mathbf{W}}$, $s > 0$ is a coefficient which controls the sharpness of the output curve, $\Delta$ determines the width of its zero-output region, and the denominator $2 \tanh(s)$ controls the output boundary. This transition function has several appealing properties: (1) for any choice of $s$, it has a symmetric output curve about the origin and is differentiable in terms of $W$; (2) when $s \to 0$, it approximates the identity mapping $f(\cdot) = W$; (3) when $s$ increases from a small positive value to a sufficiently large value with a uniform step size, it behaves as a finite sequence of continuous mappings with progressively increasing sharpness, and asymptotically converges toward the ternary output in a differentiable manner. Empirically, when $s = 4.95$, the output is already scaled within $[-1, 1]$; (4) when $s \to \infty$, it approaches the hard ternarization defined in Equation 2. We couple this transition function with the LM component, and adapt them across calibration learn-

ing iterations, forming our softened ternarization process:

$$T_i = \begin{cases} W_i, & if \ t = 0 \\ f(\hat{W}_i; \frac{t}{\gamma} s_0, \delta_\Delta \Delta_0), & if \ 0 < t \leq \gamma \\ Q(\hat{W}_i; \delta_\Delta \Delta_0), & if \ \gamma < t \leq 1. \end{cases} \quad (6)$$

Let $m$ denote the total number of calibration epochs (by default, we set $m = 60$ when using 512 calibration samples) for ternarizing the weights of a pre-trained LLM. Here, $\gamma$ denotes the calibration epoch ratio allocated for differentiable ternarization based on a finite sequence of continuous mappings, $1 - \gamma$ denotes the calibration epoch ratio allocated for hard ternarization, $t$ denotes the normalized time state of the current calibration epoch, and $s_0 > 0$ is a constant to initialize the sharpness (by default, we set $s_0 = 30$) of the transition function $f(\cdot)$. Specifically, for a linear layer, taking its pre-trained high-precision weights $\mathbf{W}$ as the starting point, the ST process undergoes two stages. In the first stage, ST produces an asymptotic ternary output by performing continuous quantization from the identity mapping to differentiable ternarization of the transformed weights $\hat{\mathbf{W}}$ via a finite sequence of continuous mappings, which is governed by the transition function $f(\cdot)$ with progressively increasing sharpness along the normalized calibration time state $0 < t \leq \gamma$. Intriguingly, the asymptotic ternary output from the first stage closely approximates a hard ternarization solution. Naturally, in the second stage, ST proceeds with hard ternarization to get a final solution, taking the gradients computed in the last iteration of the first stage for subsequent updates of three learnable factors. By this two-stage relay of differentiable ternarization and hard ternarization, ST establishes a smooth transition that effectively guides the ternarization process toward stable convergence. Figure 3 shows an illustration of the ST process.

### 2.4. Sliding-Layer Ternarization Optimization

In implementation, we adopt a sliding-layer output reconstruction scheme instead of the predominant layer-wise weight reconstruction used in existing LLM ternarization methods. We are inspired by recent works (Liu et al., 2024b; Ding et al., 2025; Wang et al., 2026) that show quantizing multiple layers together tends to yield reduced quantization errors against quantizing a single layer as it makes neighboring layers be aware of each other, enabling to use layer dependencies to mitigate information loss. However, to the best of our knowledge, this methodology has not yet been explored for ternarizing LLMs. Motivated by this, we combine CAT-Q with the framework of SliderQuant (Wang et al., 2026) to form our output reconstruction objective for ternarizing the weights of LLMs in the PTQ regime.

Let $\mathcal{W} = \{\mathbf{W}_1, ... \mathbf{W}_l\}$ denote the set of high-precision weights for the current sliding window consisting of $l$ layers in a pre-trained LLM, and let $\mathbf{X}$ denote its input feature

corresponding to a small set of calibration samples. Then, the optimization objective of our CAT-Q is to minimize an $L_2$-normed loss function defined as:

$$\underset{\mathcal{A}, \mathcal{T}}{argmin} \ ||\mathcal{F}(\mathcal{W}, \mathbf{X}) - \mathcal{F}(\mathcal{A} \cdot \mathcal{T}, \mathbf{X})||_2^2. \quad (7)$$

Here, $\mathcal{F}(\cdot, \cdot)$ denotes the output feature of the current sliding window, $\mathcal{A} = \{\alpha_1, ..., \alpha_l\}$ and $\mathcal{T} = \{\mathbf{T}_1, ..., \mathbf{T}_l\}$ denote the set of scaling factors and the set of ternary weights to be solved via the softened ternarization defined in Equation 6, and $\mathcal{A} \cdot \mathcal{T} = \{\alpha_1 \mathbf{T}_1, ..., \alpha_l \mathbf{T}_l\}$. During optimization, CAT-Q forces the window-wise outputs computed with ternary weights to match those computed with high-precision weights under the same calibration inputs. This relaxed formulation induces an implicit weight reconstruction, thereby further alleviating the difficulty of optimization.

## 3. Experiments

In this section, we conduct extensive experiments to validate the efficacy of our method, compare it with lots of related methods, and analyze the effect of key design choices.

### 3.1. Setup

To ensure a comprehensive evaluation, we apply our method to a diverse set of LLMs, varying in architecture and model size. Specifically, we select 7 dense models from the Llama2 and Qwen3 families (Touvron et al., 2023; Yang et al., 2025), as well as 3 sparsely-gated mixture of experts (MoE) models including Qwen3-30B-A3B, Qwen3-235B-A22B, and Ring-flash-2.0 (100B-A6.1B) (Team et al., 2025), covering a wide model size range from 1.7B to 235B parameters. By default, we use 512 samples randomly selected from C4 (Raffel et al., 2020) for calibration, each with a length of 2048 tokens. As for the choice of sliding window size, our method follows the default setting of SliderQuant (Wang et al., 2026). Following common practices in QAT-based ternarization research, all models are primarily evaluated in a zero-shot setting on five widely adopted commonsense reasoning benchmarks: PIQA (Bisk et al., 2020), ARC-Easy (ARC-e) and ARC-Challenge (ARC-c) (Clark et al., 2018), HellaSwag (HS) (Zellers et al., 2019), and Winogrande (WG) (Sakaguchi et al., 2021).

### 3.2. Counterpart Methods

We show the advantages of CAT-Q, our proposed 1.58-bit PTQ method for LLMs, through a two-part comparison. First, we compare CAT-Q with existing state-of-the-art 1.58-bit quantization methods that rely on quantization-aware training (QAT), including BitNet 1.58-bit v1 (Ma et al., 2024), BitNet 1.58-bit v2 (Wang et al., 2025), TernaryLLM (Chen et al., 2024b), TriLM (Kaushal et al., 2025) and Tequila (Huang et al., 2026). Second, we compare

*Table 1.* Performance of different ternary LLMs produced by CAT-Q in the PTQ regime and evaluated on five zero-shot commonsense reasoning benchmarks. Qwen3-30B-A3B, Ring-flash-2.0 (RF2-100B-A6.1B) and Qwen3-235B-A22B are MoE models, while the others are dense models. The metric is accuracy (%).

| Method | #Bits | PIQA↑ | ARC-e↑ | ARC-c↑ | HS↑ | WG↑ | Avg↑ |
|---|---|---|---|---|---|---|---|
| Qwen3-1.7B | W16A16 | 72.25 | 69.95 | 42.66 | 60.36 | 61.88 | 61.42 |
| + CAT-Q | W1.58A8 | 68.16 | 54.94 | 28.67 | 47.22 | 54.62 | 50.72 |
| + CAT-Q | W1.58A16 | 68.40 | 55.88 | 28.16 | 47.77 | 54.83 | 51.01 |
| Qwen3-4B | W16A16 | 75.19 | 78.32 | 53.84 | 68.41 | 65.51 | 68.25 |
| + CAT-Q | W1.58A8 | 68.65 | 62.64 | 35.69 | 51.48 | 63.51 | 56.39 |
| + CAT-Q | W1.58A16 | 70.62 | 62.29 | 36.95 | 53.24 | 62.19 | 57.06 |
| Qwen3-8B | W16A16 | 77.42 | 80.93 | 56.57 | 74.92 | 68.03 | 71.57 |
| + CAT-Q | W1.58A8 | 72.09 | 68.94 | 42.49 | 58.64 | 62.67 | 60.96 |
| + CAT-Q | W1.58A16 | 73.45 | 69.65 | 40.70 | 59.57 | 65.43 | 61.76 |
| Qwen3-14B | W16A16 | 79.65 | 82.83 | 60.58 | 78.83 | 73.16 | 75.01 |
| + CAT-Q | W1.58A16 | 76.22 | 72.77 | 45.22 | 67.19 | 68.03 | 65.88 |
| Qwen3-32B | W16A16 | 81.88 | 83.12 | 60.92 | 82.66 | 73.56 | 76.43 |
| + CAT-Q | W1.58A16 | 77.91 | 78.58 | 53.41 | 74.33 | 71.74 | 71.19 |
| Llama2-7B | W16A16 | 78.84 | 74.62 | 46.42 | 75.90 | 69.46 | 69.04 |
| + CAT-Q | W1.58A16 | 72.91 | 60.06 | 33.62 | 60.95 | 60.93 | 57.69 |
| Llama2-70B | W16A16 | 82.81 | 80.85 | 57.59 | 83.86 | 77.58 | 76.53 |
| + CAT-Q | W1.58A16 | 80.34 | 80.13 | 53.24 | 74.56 | 75.34 | 72.72 |
| Qwen3-30B-A3B | W16A16 | 80.14 | 79.25 | 56.23 | 77.66 | 69.93 | 72.64 |
| + CAT-Q | W1.58A16 | 74.97 | 67.55 | 43.17 | 63.22 | 63.85 | 62.55 |
| RF2-100B-A6.1B | W16A16 | 81.83 | 81.99 | 61.01 | 79.27 | 70.32 | 74.88 |
| + CAT-Q | W1.58A16 | 76.50 | 70.45 | 45.05 | 67.46 | 63.93 | 64.68 |
| Qwen3-235B-A22B | W16A16 | 82.92 | 84.89 | 65.27 | 85.87 | 79.01 | 79.59 |
| + CAT-Q | W1.58A16 | 79.16 | 74.20 | 48.21 | 74.02 | 69.85 | 69.09 |

*Table 2.* Comparison of CAT-Q and different QAT-based ternarization methods under W1.58A16 quantization. #Tokens denotes the number of training tokens for quantization.

| Model | #Tokens | PIQA↑ | ARC-e↑ | ARC-c↑ | HS↑ | WG↑ | Avg↑ |
|---|---|---|---|---|---|---|---|
| BitNetV1-1.3B | 100B | 68.80 | 54.90 | 24.20 | 37.70 | 55.80 | 48.28 |
| TriLM-1.5B | 300B | 70.30 | 59.00 | 24.70 | 40.90 | 56.10 | 50.20 |
| Qwen3-1.7B + CAT-Q | 1M | 68.40 | 55.88 | 28.16 | 47.77 | 54.83 | **51.01** |
| BitNetV1-3.9B | 100B | 73.20 | 64.20 | 28.70 | 44.20 | 60.50 | 54.16 |
| TriLM-3.9B | 300B | 74.44 | 66.00 | 31.90 | 48.30 | 62.10 | 56.55 |
| TequilaLLM-3B | 10B | 73.90 | 70.20 | 34.60 | 46.40 | 62.70 | 57.60 |
| Qwen3-4B + CAT-Q | 1M | 70.62 | 62.29 | 36.95 | 53.24 | 62.19 | 57.06 |
| Qwen3-4B + CAT-Q | 2M | 71.55 | 66.41 | 39.08 | 55.01 | 61.96 | **58.80** |
| BitNetV1-7B | 100B | 74.37 | 59.51 | 31.74 | 61.49 | 59.98 | 57.42 |
| TernaryLLM-8B+KD | 1T | 73.70 | 61.20 | 36.40 | 63.90 | 65.00 | 60.04 |
| Qwen3-8B + CAT-Q | 1M | 73.45 | 69.65 | 40.70 | 59.57 | 65.43 | **61.76** |

CAT-Q with some recent PTQ methods under their ultra-low-bit configurations: 2-bit methods including GPTQ (Frantar et al., 2023), AWQ (Lin et al., 2024b), OmniQuant (Shao et al., 2024), EfficientQAT (Chen et al., 2025) and SliderQuant (Wang et al., 2026), as well as dual 1-bit methods including BiLLM (Huang et al., 2024), PB-LLM (Shang et al., 2024) and DB-LLM (Chen et al., 2024a).

### 3.3. Main Results

**Ternary Quantization Results of CAT-Q.** Table 1 summarizes the results of CAT-Q across 10 pre-trained LLMs, covering diverse model architectures (dense and MoE), sizes (1.7B to 235B parameters) and families (Qwen3, Llama2, and Ring-flash-2.0). The performance degradation caused by quantization decreases as the model size increases, with

*Table 3.* Comparison of CAT-Q and different QAT-based ternarization methods under W1.58A8 quantization. #Tokens denotes the number of training tokens for quantization.

| Model | #Tokens | PIQA↑ | ARC-e↑ | ARC-c↑ | HS↑ | WG↑ | Avg↑ |
|---|---|---|---|---|---|---|---|
| BitNetV2-1.3B | 100B | 69.42 | 49.96 | 27.90 | 48.37 | 57.22 | 50.57 |
| Qwen3-1.7B + CAT-Q | 1M | 68.16 | 54.94 | 28.67 | 47.22 | 54.62 | **50.72** |
| BitNetV2-3B | 100B | 71.33 | 55.56 | 30.55 | 57.19 | 58.72 | 54.67 |
| Qwen3-4B + CAT-Q | 1M | 68.65 | 62.64 | 35.69 | 51.48 | 63.51 | **56.39** |
| BitNetV2-7B | 100B | 74.10 | 58.54 | 32.94 | 61.08 | 61.48 | 57.63 |
| Qwen3-8B + CAT-Q | 1M | 72.09 | 68.94 | 42.49 | 58.64 | 62.67 | **60.96** |

*Table 4.* Comparison of CAT-Q and recent PTQ methods with bit-widths close to 1.58-bit. PB-LLM[†], DB-LLM[†] and BiLLM[†] use dual 1-bit representations (or say, binary residual) and mixed-precision quantization, so W1* in them is close to or larger than W2A16 in terms of bit width. EfficientQAT[‡] is actually a PTQ method as it uses pre-trained LLMs and "QAT" is for re-training.

| Method | #Bits | PIQA↑ | ARC-e↑ | ARC-c↑ | HS↑ | WG↑ | Avg↑ |
|---|---|---|---|---|---|---|---|
| Llama2-7B | W16A16 | 78.84 | 74.62 | 46.42 | 75.90 | 69.46 | 69.04 |
| + AWQ | W2A16 | 50.00 | 26.52 | 26.79 | 26.14 | 49.64 | 35.82 |
| + GPTQ | W2A16 | 58.32 | 40.45 | 21.25 | 32.59 | 55.17 | 41.55 |
| + OmniQuant | W2A16 | 65.13 | 50.13 | 23.46 | 40.28 | 55.88 | 46.98 |
| + PB-LLM[†] | W1*A16 | 55.22 | 29.88 | 22.01 | 30.49 | 50.36 | 37.59 |
| + BiLLM[†] | W1*A16 | 60.60 | 36.20 | 24.40 | 34.80 | 52.40 | 41.68 |
| + DB-LLM[†] | W1*A16 | **73.18** | 45.20 | 33.53 | **61.98** | **61.72** | 55.12 |
| + SliderQuant | W2A16 | 70.78 | 57.79 | 31.06 | 57.15 | 60.14 | 55.38 |
| + CAT-Q | W1.58A16 | 72.91 | **60.06** | **33.62** | 60.95 | 60.93 | **57.69** |
| Llama2-70B | W16A16 | 82.81 | 80.85 | 57.59 | 83.86 | 77.58 | 76.53 |
| + GPTQ | W2A16 | 49.51 | 25.08 | 22.70 | 25.04 | 49.57 | 34.38 |
| + OmniQuant | W2A16 | 74.10 | 67.21 | 33.28 | 35.45 | 64.33 | 54.87 |
| + DB-LLM[†] | W1*A16 | 79.27 | 55.93 | 44.45 | **76.16** | 73.32 | 65.82 |
| + EfficientQAT[‡] | W2A16 | 80.20 | 80.01 | 49.23 | 61.58 | 73.64 | 68.93 |
| + SliderQuant | W2A16 | 78.79 | 77.14 | 52.71 | 73.02 | 73.75 | 71.08 |
| + CAT-Q | W1.58A16 | **80.34** | **80.13** | **53.24** | 74.56 | **75.34** | **72.72** |

Llama2-70B exhibiting only a 3.81% drop. Meanwhile, MoE models show greater quantization sensitivity compared to dense models of similar scale, likely due to their fewer activated parameters. Notably, we demonstrate for the first time that CAT-Q can quantize LLMs having up to 235B parameters into leading ternary weight models within 60 hours on 8 A100-80GB GPUs. In addition, promising results are achieved with both W1.58A8 and W1.58A16 configurations. These results highlight the broad applicability of CAT-Q across diverse models and quantization settings.

**Comparison with QAT-based Ternarization Methods.** Table 2 and Table 3 compare the model performance of CAT-Q with leading 1.58-bit QAT methods under W1.58A16 and W1.58A8 configurations. Given that these QAT methods are limited to small model scales, we ensure a fair comparison by evaluating models of similar size. We can observe that CAT-Q offers a highly accurate and cost-efficient solution for ternary quantization. For instance, on LLMs with 1.7B to 8B parameters, CAT-Q with merely 512 calibration samples achieves competitive performance to BitNet 1.58-bit v1 and v2 families trained on 100B tokens, yielding a 100,000× reduction in training tokens. Even compared to TernaryLLM-8B, which uses 1T tokens for training guided by knowledge distillation (KD), our method still attains better overall accuracy.

*Table 5.* Ablation of the complementarity of $\delta_\mu$, $\delta_\alpha$ and $\delta_\Delta$ in the learnable modulation (LM) component as well as the softened ternarization (ST) component under W1.58A16 ternarization. *The baseline in the first row corresponds to SliderQuant without the proposed LM and ST components.*

| Model | $\delta_\mu$ | $\delta_\alpha$ | $\delta_\Delta$ | ST | PIQA↑ | ARC-e↑ | ARC-c↑ | HS↑ | WG↑ | Avg↑ |
|---|---|---|---|---|---|---|---|---|---|---|
| | | | | | 55.33 | 31.99 | 26.11 | 36.70 | 50.67 | 40.16 |
| | ✓ | | | | 63.55 | 41.33 | 28.34 | 42.81 | 54.46 | 46.10 |
| | | ✓ | | | 65.61 | 51.47 | 31.23 | 44.74 | 56.59 | 49.93 |
| Qwen3-4B | ✓ | ✓ | | | 66.11 | 53.66 | 32.20 | 47.35 | 57.72 | 51.41 |
| | ✓ | ✓ | ✓ | | 68.66 | 56.95 | 33.87 | 48.30 | 58.19 | 53.19 |
| | ✓ | ✓ | | ✓ | 68.86 | 59.73 | 34.92 | 52.66 | 61.48 | 55.53 |
| | ✓ | ✓ | ✓ | ✓ | **70.62** | **62.29** | **36.95** | **53.24** | **62.19** | **57.06** |
| | | | | | 63.06 | 41.29 | 27.05 | 43.53 | 51.93 | 45.37 |
| | ✓ | | | | 67.95 | 51.77 | 28.75 | 53.21 | 56.35 | 51.61 |
| | | ✓ | | | 68.66 | 52.23 | 29.86 | 53.28 | 56.99 | 52.20 |
| Llama2-7B | ✓ | ✓ | | | 70.35 | 55.52 | 32.76 | 55.48 | 57.93 | 54.41 |
| | ✓ | ✓ | ✓ | | 72.69 | 58.98 | 31.55 | 57.85 | 58.77 | 55.97 |
| | ✓ | ✓ | | ✓ | 71.39 | 59.48 | 32.55 | 59.21 | 60.37 | 56.60 |
| | ✓ | ✓ | ✓ | ✓ | **72.91** | **60.06** | **33.62** | **60.95** | **60.93** | **57.69** |

*Table 6.* Ablation of the softened ternarization process with varying choices of the calibration epoch ratio $\gamma$ allocated for differentiable ternarization. The underlined results are for our default setting ($\gamma = 0.8$), and the best results are in bold.

| Model | $\gamma$ | PIQA↑ | ARC-e↑ | ARC-c↑ | HS↑ | WG↑ | Avg↑ |
|---|---|---|---|---|---|---|---|
| | 0.5 | 68.15 | 60.95 | 34.81 | 51.33 | 58.98 | 54.84 |
| | 0.6 | 69.04 | 61.24 | 37.75 | 51.77 | 61.80 | 55.92 |
| | 0.7 | 69.04 | 61.81 | 36.25 | 52.37 | 61.90 | 56.27 |
| Qwen3-4B | 0.8 | **70.62** | **62.29** | **36.95** | **53.24** | **62.19** | **57.06** |
| | 0.9 | 69.75 | 61.80 | 35.36 | 52.47 | 61.46 | 56.17 |
| | 1.0 | 68.77 | 59.55 | 34.98 | 51.55 | 60.27 | 55.02 |
| | 0.5 | 69.12 | 57.18 | 30.12 | 59.73 | 59.91 | 55.21 |
| | 0.6 | 70.56 | 58.81 | 31.10 | 60.13 | 60.18 | 56.16 |
| | 0.7 | 72.13 | 59.18 | 32.89 | **61.71** | 60.91 | 57.36 |
| Llama2-7B | 0.8 | **72.91** | **60.06** | **33.62** | 60.95 | 60.93 | **57.69** |
| | 0.9 | 71.56 | 59.24 | 33.13 | 59.90 | **61.48** | 57.06 |
| | 1.0 | 70.67 | 58.80 | 31.38 | 60.02 | 59.04 | 55.98 |

*Table 7.* Ablation of the softened ternarization process with varying choices of the constant $s_0$. The underlined results are for our default setting ($s_0 = 30$), and the best results are in bold.

| Model | $s_0$ | PIQA↑ | ARC-e↑ | ARC-c↑ | HS↑ | WG↑ | Avg↑ |
|---|---|---|---|---|---|---|---|
| | 10 | 69.42 | 59.68 | 35.32 | 51.34 | 59.04 | 54.96 |
| | 15 | 69.61 | 60.72 | 35.79 | 52.53 | 60.83 | 55.90 |
| Qwen3-4B | 30 | **70.62** | **62.29** | **36.95** | 53.24 | **62.19** | **57.06** |
| | 100 | 69.26 | 61.24 | 35.75 | **53.33** | 61.48 | 56.21 |
| | 200 | 68.63 | 58.07 | 33.19 | 48.27 | 58.96 | 53.42 |
| | 1000 | 66.59 | 53.28 | 31.31 | 45.48 | 57.46 | 50.82 |
| | 10 | 70.46 | 54.55 | 31.48 | 57.57 | 56.51 | 54.11 |
| | 15 | 71.28 | 56.88 | 32.41 | 59.12 | 58.46 | 55.63 |
| Llama2-7B | 30 | **72.91** | **60.06** | **33.62** | **60.95** | **60.93** | **57.69** |
| | 100 | 72.42 | 59.46 | 33.14 | 59.94 | 59.51 | 56.89 |
| | 200 | 70.72 | 56.33 | 31.65 | 57.43 | 57.41 | 54.71 |
| | 1000 | 66.65 | 45.88 | 27.13 | 50.11 | 56.12 | 49.18 |

**Role of Key Designs.** We first perform an ablation to evaluate the complementarity of our two core components: LM and ST. Table 5 summarizes the results. Recall that LM leverages a composition of three learnable factors $\delta_\mu$, $\delta_\alpha$, and $\delta_\Delta$ to mitigate the distributional weight misalignment and the ternary threshold. Progressively incorporating these factors yields increasing performance gains, and the addition of ST significantly boosts the performance gain, validating the importance of the transition from differentiable ternarization to hard ternarization. Coupling LM and ST leads to the best accuracy on both Qwen3-4B and Llama2-7B, confirming the effectiveness and synergy of our core designs.

**Choices of Calibration Epoch Ratio $\gamma$.** The ratio $\gamma$ controls the allocation of calibration epochs between the two stages of ST, namely a differentiable ternarization stage and a following hard ternarization stage. As shown in Table 6, CAT-Q exhibits robust ternarization performance across all settings with $\gamma \geq 0.5$. A sweet spot is achieved at $\gamma = 0.8$, which we set as the default. Empirically, we recommend setting $\gamma$ between 0.8 and 0.9. For ST, this range provides sufficient epochs for a smooth transition from the identity mapping to differentiable ternarization in the first stage while reserving adequate epochs for converging to a good hard ternarization solution in the second stage. Although $\gamma = 1.0$ remains viable, it leads to a relative accuracy drop, confirming the necessity of the hard ternarization stage.

**Choices of Constant $s_0$.** The constant $s_0$ controls the maximum sharpness reached at the end of the differentiable ternarization stage in ST. As shown in Table 7, CAT-Q achieves the best average performance with $s_0 = 30$ on both Qwen3-4B and Llama2-7B, which we set as the default. Moderate values of $s_0$, such as 30 and 100, provide a good balance between smooth optimization and sufficient approximation to hard ternarization. In contrast, a small $s_0$ keeps the softened ternarization overly smooth and away from the target hard behavior, while an excessively large $s_0$

**Comparison with Recent PTQ Methods.** In Table 4, CAT-Q is further compared with state-of-the-art PTQ methods under their ultra-low-bit settings, which consist of 2-bit methods and dual 1-bit methods (whose actual bit-widths are around 2-bit). Despite operating at a lower 1.58-bit weight representation, in terms of the average accuracy, CAT-Q is superior to these methods across both 7B and 70B model scales. These results demonstrate that CAT-Q achieves a substantially better efficiency-accuracy trade-off, providing larger compression and lower computational cost while maintaining better performance mostly.

### 3.4. Ablation Studies

To have a better understanding of the components and designs of CAT-Q, we perform a series of ablative experiments.

*Table 8.* Ablation of different strategies to determine the scaling factor $\alpha$ and the threshold $\Delta$. We compare three strategies analyzed in the Method section, including static approximation (SA), direct learning (DL) and our learnable modulation (LM). The underlined results are for our default setting, and the best results are in bold.

| Model | Estimating Strategy | PIQA↑ | ARC-e↑ | ARC-c↑ | HS↑ | WG↑ | Avg↑ |
|---|---|---|---|---|---|---|---|
| Qwen3-4B | SA | 55.33 | 31.99 | 26.11 | 36.70 | 50.67 | 40.16 |
| | DL | 57.90 | 53.38 | 26.79 | 46.14 | 50.51 | 46.94 |
| | LM (ours) | **70.62** | **62.29** | **36.95** | **53.24** | **62.19** | **57.06** |
| Llama2-7B | SA | 63.06 | 41.29 | 27.05 | 43.53 | 51.93 | 45.37 |
| | DL | 67.15 | 51.61 | 29.73 | 45.46 | 57.47 | 50.23 |
| | LM (ours) | **72.91** | **60.06** | **33.62** | **60.95** | **60.93** | **57.69** |

*Table 9.* Ablation of LM with vs. without $\mu$ for weight reconstruction namely $\mathbf{W} \approx \alpha\mathbf{T} + \mu$ vs. $\mathbf{W} \approx \alpha\mathbf{T}$. The underlined results are for our default setting, and the best results are in bold.

| Model | Weight Reconstruction | PIQA↑ | ARC-e↑ | ARC-c↑ | HS↑ | WG↑ | Avg↑ |
|---|---|---|---|---|---|---|---|
| Qwen3-4B | with $\mu$ | 69.80 | 61.53 | **36.95** | **53.98** | 60.69 | 56.59 |
| | without $\mu$ | **70.62** | **62.29** | **36.95** | 53.24 | **62.19** | **57.06** |
| Llama2-7B | with $\mu$ | **74.18** | **60.26** | **34.13** | 58.01 | 59.41 | 57.20 |
| | without $\mu$ | 72.91 | 60.06 | 33.62 | **60.95** | **60.93** | **57.69** |

*Table 10.* Ablation of CAT-Q with learnable vs. different hand-crafted choices of the threshold $\Delta$. The underlined results are for our default setting, and the best results are in bold.

| Model | $\Delta$ | PIQA↑ | ARC-e↑ | ARC-c↑ | HS↑ | WG↑ | Avg↑ |
|---|---|---|---|---|---|---|---|
| Qwen3-4B | 0.125 | 64.96 | 51.14 | 29.18 | 43.84 | 55.64 | 48.35 |
| | 0.250 | 67.57 | 54.63 | 32.85 | 48.66 | 58.09 | 52.76 |
| | 0.500 | 68.86 | 59.73 | 34.92 | 52.66 | 61.48 | 55.53 |
| | 1.000 | 64.69 | 45.29 | 27.47 | 41.16 | 54.62 | 46.65 |
| | 2.000 | 56.47 | 34.05 | 21.16 | 28.93 | 49.64 | 38.45 |
| | Learnable | **70.62** | **62.29** | **36.95** | **53.24** | **62.19** | **57.06** |
| Llama2-7B | 0.125 | 60.12 | 35.98 | 24.40 | 34.98 | 53.04 | 41.30 |
| | 0.250 | 71.27 | 55.09 | 29.78 | 46.06 | 58.09 | 52.06 |
| | 0.500 | 71.39 | 59.48 | 32.55 | 59.21 | 60.37 | 56.60 |
| | 1.000 | 62.51 | 40.87 | 26.71 | 36.73 | 52.72 | 43.51 |
| | 2.000 | 55.54 | 31.90 | 24.70 | 30.25 | 51.07 | 38.29 |
| | Learnable | **72.91** | **60.06** | **33.62** | **60.95** | **60.93** | **57.69** |

makes the transition reach the hard target too early and weakens the benefit of differentiable optimization. This suggests that ST requires a gradual yet sufficiently sharp transition, and we recommend setting $s_0$ around 30 in practice.

**Strategies to Determine $\alpha$ and $\Delta$.** Note the LM component mitigates the distributional misalignment by adaptively transforming the weight distribution with learnable factors $\delta_\mu$, $\delta_\alpha$, and $\delta_\Delta$, making weights less sensitive to ternarization. As evidenced by the lower weight reconstruction errors shown in Figure 2, compared to SA and DL, LM better aligns the reconstructed weight distribution with the original weights. The comparative results in Table 8 further validate this conclusion, where LM consistently achieves the best results, underscoring its importance within our CAT-Q.

**Weight Reconstruction Strategies.** In LM, $\mu$ is used to

*Table 11.* Ablation of CAT-Q with varying numbers of calibration samples. The underlined results are for our default setting, and the best results are in bold.

| Model | #Samples | PIQA↑ | ARC-e↑ | ARC-c↑ | HS↑ | WG↑ | Avg↑ |
|---|---|---|---|---|---|---|---|
| Qwen3-4B | 32 | 64.15 | 50.51 | 28.75 | 44.30 | 56.43 | 48.43 |
| | 64 | 66.65 | 53.45 | 31.57 | 47.95 | 60.62 | 52.85 |
| | 128 | 68.34 | 59.51 | 34.64 | 50.12 | 59.04 | 54.73 |
| | 256 | 68.02 | 60.19 | 35.43 | 51.34 | 59.09 | 54.81 |
| | 512 | 70.62 | 62.29 | 36.95 | 53.24 | **62.19** | 57.06 |
| | 1024 | **71.55** | **66.41** | **39.08** | **55.01** | 61.96 | **58.80** |
| Llama2-7B | 32 | 67.03 | 47.65 | 24.66 | 49.77 | 53.09 | 48.44 |
| | 64 | 67.85 | 51.63 | 28.50 | 51.31 | 55.96 | 51.85 |
| | 128 | 69.53 | 55.03 | 29.33 | 56.10 | 58.56 | 53.71 |
| | 256 | 71.00 | 56.51 | 32.51 | 58.02 | 58.70 | 55.75 |
| | 512 | 72.91 | 60.06 | 33.62 | 60.95 | 60.93 | 57.69 |
| | 1024 | **74.25** | **63.07** | **36.29** | **63.12** | **63.75** | **60.50** |

*Table 12.* Ablation of CAT-Q with varying quantization group sizes. The underlined results are for our default setting, and the best results are in bold.

| Model | Group Size | PIQA↑ | ARC-e↑ | ARC-c↑ | HS↑ | WG↑ | Avg↑ |
|---|---|---|---|---|---|---|---|
| Qwen3-4B | 64 | **70.65** | **62.39** | **37.82** | **54.18** | 61.22 | **57.25** |
| | 128 | 70.62 | 62.29 | 36.95 | 53.24 | **62.19** | 57.06 |
| | 256 | 69.48 | 61.49 | 36.43 | 52.97 | 60.06 | 56.09 |
| | 512 | 69.01 | 61.03 | 35.78 | 51.98 | 59.64 | 55.49 |
| | Channel-wise | 68.85 | 60.73 | 35.19 | 51.13 | 59.41 | 55.06 |
| Llama2-7B | 64 | **73.18** | **61.45** | **34.48** | **61.35** | 60.04 | **58.10** |
| | 128 | 72.91 | 60.06 | 33.62 | 60.95 | **60.93** | 57.69 |
| | 256 | 72.09 | 59.45 | 32.89 | 59.79 | 59.83 | 56.81 |
| | Channel-wise | 71.49 | 58.57 | 31.55 | 59.47 | 59.75 | 56.17 |

redistribute the weights from $\mathbf{W}$ to $\hat{\mathbf{W}}$ but omitted in the reconstruction $\mathbf{W} \approx \alpha\mathbf{T}$ for optimization. As shown in Table 9, this simplified form achieves slightly improved accuracy over retaining $\mu$ ($\mathbf{W} \approx \alpha\mathbf{T}+\mu$), while maintaining the hardware-friendly deployment consistent with TWN.

**Learnable Threshold vs. Hand-crafted Thresholds.** In LM, the threshold $\Delta$ is optimized as $\Delta = \delta_\Delta \Delta_0$ where $\delta_\Delta$ is a learnable factor. We compare this learnable design against several hand-crafted thresholds in Table 10. The results confirm that the learnable $\Delta$ achieves the best performance, validating its effectiveness within our LM component.

**Effect of the Number of Calibration Samples.** We also ablate the effect of the calibration sample count in Table 11. The accuracy improves consistently as the sample count increases: on both Qwen3-4B and Llama2-7B, CAT-Q attains the best performance when using 1024 samples. We employ 512 samples as our default setting, which makes a favorable balance between ternarization cost and model accuracy.

**Effect of Group Size.** We finally ablate the impact of the quantization group size in Table 12. The results indicate that smaller group sizes generally lead to better performance. We use the group size of 128 as default, following recent 1.58-bit QAT-based works TriLM and Tequila for fair comparisons with them.

*Table 13.* Comparison of model deployment cases across different low-bit formats with `llama.cpp` on an Intel Core i9-13900KF CPU and an NVIDIA GeForce RTX 4090 GPU. We report profiling results on both CPU and GPU for standard 4-bit and 2-bit models, together with the 1.58-bit model obtained by CAT-Q, which is implemented as `TQ1_0` on CPU and `TQ2_0` on GPU. We set the batch size to 1 and the generation length to 512 tokens.

| Model | Deployment Format | Weight Memory (GB) | CPU \| GPU Throughput (Tokens/s) |
|---|---|---|---|
| Qwen3-4B | W4A16 (Q4_K_M) | 2.32 | 19.31 \| 225.91 |
| | W2A16 (Q2_K_M) | 1.55 | 23.84 \| 268.51 |
| | CAT-Q (TQ1_0 \| TQ2_0) | **1.01** \| **1.17** | **37.35** \| **320.34** |
| Llama2-7B | W4A16 (Q4_K_M) | 3.80 | 11.83 \| 174.34 |
| | W2A16 (Q2_K_M) | 2.63 | 17.01 \| 210.72 |
| | CAT-Q (TQ1_0 \| TQ2_0) | **1.62** \| **1.90** | **26.30** \| **281.66** |

**Acceleration in Model Deployment.** We further deploy ternary LLMs produced by CAT-Q in `llama.cpp` using its ternary formats, i.e., `TQ1_0` on CPU and `TQ2_0` on GPU. As shown in Table 13, our ternary models consistently achieve lower weight memory and higher decoding throughput than standard 4-bit and 2-bit formats on both Qwen3-4B and Llama2-7B. The efficiency gains hold on both the Intel Core i9-13900KF CPU and the NVIDIA GeForce RTX 4090 GPU, showing that CAT-Q can translate its 1.58-bit representation into practical inference efficiency improvements.

**In the Appendix**, we further provide: (1) a summary of benchmark datasets and implementation details; (2) illustrations of the smooth transition function $f(\cdot)$ with varying sharpness choices of $s$; (3) a pilot study of CAT-Q on challenging mathematics and coding tasks; (4) a comparison of loss curves of CAT-Q with vs. without ST; (5) a more comprehensive comparison of different strategies for determining $\alpha$ and $\Delta$; (6) a discussion of limitations.

## 4. Related Work

Beyond the methods discussed earlier, in this section, we briefly review other relevant PTQ methods.

There exist numerous PTQ methods (Zhao et al., 2019; Banner et al., 2019; Nagel et al., 2020; Li et al., 2021; Wei et al., 2022a) that focus on convolutional neural networks for computer vision tasks. In contrast, LLMs have substantially larger model sizes and are harder to quantize, mainly due to the presence of outlier elements (a small fraction of salient weights/activations with magnitudes significantly larger than the rest) which induce severe quantization errors. To tackle this problem, a variety of PTQ methods are proposed, among which mixed-precision quantization is a widely adopted scheme. It's basic idea is to isolate outliers in high-precision format (e.g., FP16) and quantize the remaining weights/activations into low-precision representations (e.g., INT8/INT4). Representative examples include Q-BERT (Shen et al., 2020), LLM.int8() (Dettmers et al.,

2022), SpQR (Dettmers et al., 2024), QUIK (Ashkboos et al., 2024a) and SqueezeLLM (Kim et al., 2024). However, they are not hardware-efficient for deployment. Instead, Outlier Suppression (Wei et al., 2022b) combines LayerNorm migration and token-wise clipping to make activations more amenable to 8-bit quantization. ZeroQuant (Yao et al., 2022) adopts a fine-grained INT8 quantization scheme consisting of group-wise quantization for weights and token-wise quantization for activations. GPTQ (Frantar et al., 2023) leverages approximate second-order Hessian matrices to suppress the impact of weight outliers. Another line of PTQ research for LLMs relies on equivalent transformations to mitigate the quantization difficulty of outliers. For instance, SmoothQuant (Xiao et al., 2023) applies channel-wise smoothing, QLLM (Liu et al., 2024b) adopts channel-split-merge, QuIP (Chee et al., 2024) employs incoherence processing and QuaRot (Ashkboos et al., 2024b) uses random rotation matrices. Subsequent variants improve them by learning transformations (Shao et al., 2024; Lin et al., 2024b;a; Tseng et al., 2024; Liu et al., 2025a; Sun et al., 2025). These PTQ methods primarily consider basic low-bit settings and perform poorly in ultra-low-bit settings (e.g., 3-bit quantization and 2-bit quantization). Our method differs with them in focus, formulation and deployment.

There also exist some works (Huang et al., 2024; Shang et al., 2024; Li et al., 2025) tailored for binary quantization of LLMs. As another standalone research direction, binary quantization is more challenging than the ternary quantization (this paper's focus). In formulation, these methods usually adopt mixed-precision quantization and binary residuals (dual 1-bit representations) to avoid performance collapse, following earlier binarization techniques developed for vision models (Lin et al., 2017; Guo et al., 2017).

## 5. Conclusion

In this paper, we introduce CAT-Q, a cost-efficient and accurate quantization method for ternarizing the weights of LLMs in the post-training regime. It consists of two core components, learnable modulation and softened ternarization, which are coupled into a sliding-layer quantization pipeline to learn scaling factors and ternary thresholds across multiple layers jointly. Extensive experiments show the advantages of CAT-Q over existing methods that rely on QAT. Our method makes it substantially easier to obtain and deploy high-performing ternary weight LLMs in practice.

## Impact Statement

This paper presents work whose goal is to advance the field of machine learning. There are many potential societal consequences of our work, none of which we feel must be specifically highlighted here.

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

# Appendix

## A. Datasets Used in Experiments

**C4** (Raffel et al., 2020) (Colossal Clean Crawled Corpus) is a large-scale corpus widely adopted for language modeling research. It consists of approximately 156 billion tokens collected from web documents and processed through extensive filtering and cleaning procedures based on Common Crawl.

**PIQA** (Bisk et al., 2020) is a benchmark for physical commonsense reasoning, comprising about 16,000 training instances and 3,000 validation instances. Each example is presented as a multiple-choice question with two candidate solutions, where exactly one option is correct.

**ARC** (Clark et al., 2018) contains 7,787 multiple-choice science questions at the grade-school level, designed to promote progress in question answering and reasoning. The dataset is divided into an Easy set and a Challenge set, with the latter including questions that are not solvable by simple retrieval or co-occurrence–based methods.

**HellaSwag** (Zellers et al., 2019) includes roughly 70,000 training samples and 10,000 validation samples. It evaluates commonsense reasoning by requiring models to select the most plausible continuation of a given context, which is constructed from crowdsourced activity descriptions and image captions.

**Winogrande** (Sakaguchi et al., 2021) consists of approximately 44,000 instances formulated as binary-choice fill-in-the-blank problems. The task requires selecting the correct option to complete a sentence, emphasizing coreference resolution and commonsense reasoning capabilities.

**MATH-500** (Hendrycks et al., 2021) comprises 500 challenging competition-level mathematics problems sampled from the MATH dataset. These problems span various topics such as algebra, geometry, number theory, and probability, and are designed to test a model's ability to perform complex mathematical reasoning and generate step-by-step solutions with rigorous intermediate derivations.

**Omni-MATH** (Gao et al., 2025) is a large-scale benchmark for olympiad-level mathematical reasoning, containing 4,428 problems collected from diverse national and international mathematics competitions. The dataset covers a broad range of advanced topics and is designed to evaluate models on difficult multi-step problem solving beyond standard school-level math benchmarks, posing higher demands on abstraction, symbolic manipulation, and long-chain reasoning.

**GSM8K** (Cobbe et al., 2021) is a dataset of 8,792 high-quality, linguistically diverse grade school math word problems created by human problem writers. The dataset is segmented into 7,473 training problems and 1,319 test problems, each requiring multi-step reasoning and basic arithmetic operations to solve, while emphasizing the ability to translate natural language descriptions into executable reasoning procedures.

**HumanEval+** (Liu et al., 2023) is an extension of the HumanEval dataset, consisting of 164 original programming problems designed to assess the functional correctness of code generated by language models. Each problem includes a function signature, a docstring specifying the intended functionality, and multiple test cases for evaluation, enabling a more reliable assessment of whether generated programs satisfy the expected behavior.

**MBPP+** (Liu et al., 2023) is an augmented version of the Mostly Basic Programming Problems (MBPP) dataset, comprising approximately 378 crowd-sourced Python programming tasks. Each task includes a natural language description, a reference solution, and three test cases, aiming to evaluate models' abilities in basic programming and problem-solving across diverse everyday coding scenarios.

# B. Implementation Details of CAT-Q

## B.1. Hyper-parameter Settings

Table A summarizes the complete hyper-parameter configuration used in our experiments. We perform group-wise ternary quantization with a fixed group size of 128 and use a calibration set constructed from C4, consisting of 512 samples with 2048 tokens per sample. The table reports the initialization, including the ternary threshold $\Delta_0 = 0.5$, the constant $s_0 = 30$ and the differentiable ternarization ratio $\gamma = 0.8$. It also summarizes the optimization-related settings of the batch size, the optimizer, the number of training epochs, and the learning rate schedule. Unless otherwise stated, all experiments follow this configuration.

*Table A.* The detailed hyper-parameter settings of CAT-Q.

| Configuration | Setting |
|---|---|
| Calibration set | C4 |
| Number of calibration samples | 512 |
| Tokens per sample | 2048 |
| Group size | 128 |
| $\Delta_0$ | 0.5 |
| $s_0$ | 30 |
| $\gamma$ | 0.8 |
| Batch size | 3 |
| Optimizer | AdamW |
| Epochs | 60 |
| Learning rate of learnable modulation | 0.001 |
| Learning rate schedule | linear decay to zero |

## B.2. Quantization Details

We follow the ternary quantization formulation introduced in the main text and do not restate the derivation here. This section only summarizes implementation details that are not explicitly covered in the main paper. In practice, we adopt a group-wise quantization scheme, where the weight tensor is flattened and partitioned into non-overlapping groups with a fixed group size of $g = 128$, and all related statistics and parameters are handled independently at the group level. After group-wise normalization and ternarization as described in the main paper, the original weights are approximated as $\mathbf{W} \approx \alpha\mathbf{T}$ without $\mu$. At deployment time, the ternary weights remain in $\{-1, 0, +1\}$, and only the group-wise scaling factor $\alpha$ is required, which can be absorbed into surrounding linear operations. This zero-point–free design preserves the hardware-friendly properties of standard ternary quantization and is applicable to both weight-only and weight–activation quantization settings. For experiments involving activation quantization, all intermediate activations are quantized following the same calibration procedure, except for the Softmax output probability vectors, which are kept in full precision for numerical stability. When handling MoE models, the routing modules used for expert selection in the MLP are excluded from quantization, as they constitute a negligible fraction of the total parameters while being critical to expert assignment.

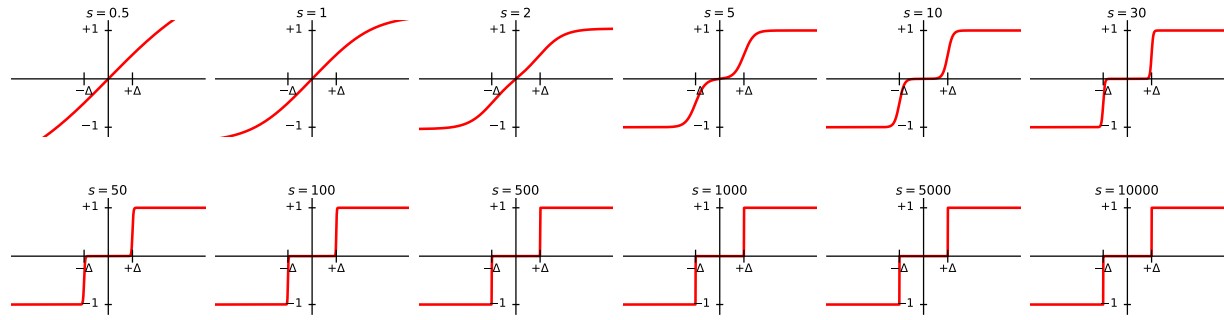

*Figure A.* Illustrative output curves of our proposed smooth transition function $f(\cdot)$ with varying sharpness choices of $s$. As $s$ increases, $f(\cdot)$ gradually transitions from a finite sequence of continuous mappings with increasing sharpness to a near-discrete ternary weight output in terms of the high-precision weight input, approximating a hard ternarization solution.

# C. Illustrations of the Smooth Transition Function $f(\cdot)$ with Varying Sharpness

Figure A visualizes the smooth transition function defined in Section 2.3 (Equation 5) under different values of the sharpness parameter $s$, illustrating how the softened ternarization state evolves as $s$ increases. Here, $s$ controls the instantaneous sharpness of the transition function during training, while $s_0$ denotes the final sharpness value reached at the end of the differentiable ternarization stage. As discussed in Section 3.4 with Table 7, an appropriate $s_0$ should make the softened ternarization sufficiently close to hard ternarization while preserving a smooth optimization trajectory. The visualization provides an intuitive explanation for this trade-off: small values of $s$ lead to an overly smooth transition, whereas excessively large values make the function nearly hard. This supports our default choice of $s_0 = 30$ in CAT-Q.

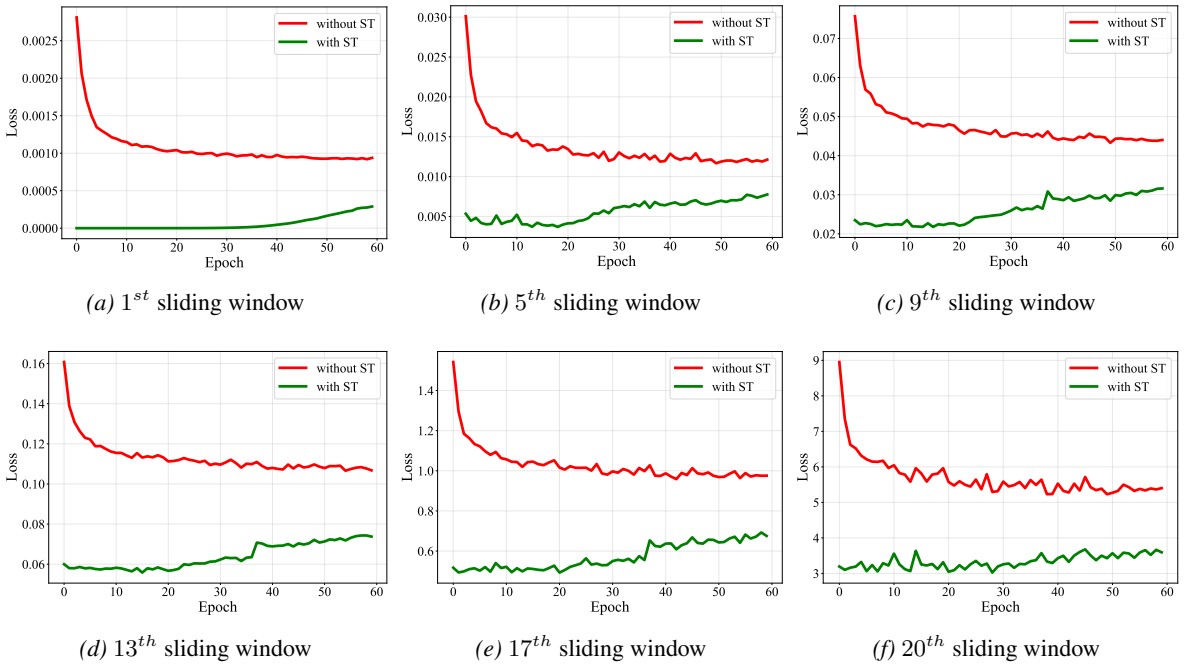

*(a) $1^{st}$ sliding window*      *(b) $5^{th}$ sliding window*      *(c) $9^{th}$ sliding window*

*(d) $13^{th}$ sliding window*      *(e) $17^{th}$ sliding window*      *(f) $20^{th}$ sliding window*

*Figure B.* Comparison of the calibration loss curves of CAT-Q with vs. without the softened ternarization (ST) component. We use the $1^{st}, 5^{th}, 9^{th}, 13^{th}, 17^{th}$ and $20^{th}$ sliding windows of our method with Qwen3-4B for illustrative comparisons. By introducing the ST component, although the loss curves of CAT-Q exhibit a different optimization trajectory trend compared to those of merely using the learnable modulation (LM) component, it consistently converges to substantially lower loss values. This behavior supports the core insight of the ST component, which enables smoother optimization and mitigates the adverse effect of early-epoch hard ternarization.

## D. Comparison of Loss Curves of CAT-Q with vs. without the ST Component

Figure B illustrates the evolution of quantization loss across training steps under the proposed sliding-layer ternarization optimization paradigm (Section 2.4). When softened ternarization (ST) is employed, the quantization loss increases smoothly during training and converges to a substantially lower final value than the baseline without ST. This behavior stems from the gradual annealing of ST from an identity-like mapping to hard ternarization, which mitigates early quantization-induced perturbations and preserves stable gradient propagation. In contrast, applying hard ternarization throughout training introduces strong non-smoothness from the outset, resulting in inferior convergence and higher residual quantization error. Overall, ST leads to smoother optimization trajectories and consistently improved convergence, validating the effectiveness of the proposed softened ternarization strategy.

## E. A Pilot Study of CAT-Q on Challenging Mathematics and Coding Tasks

We further evaluate CAT-Q on challenging mathematics and coding tasks, including Math-500 (Hendrycks et al., 2021), GSM8K (Cobbe et al., 2021), Omni-MATH (Gao et al., 2025), HumanEval+ (Liu et al., 2023), and MBPP+ (Liu et al., 2023). As shown in Table B, on these challenging tasks, directly applying CAT-Q under W1.58A16 suffers from serious performance degradation. We study the underlying causes of this issue and propose CAT-Q+ in a separate technical report to be released in the near future. CAT-Q+ introduces a new calibration data generation strategy while retaining the original CAT-Q quantization method, resulting in substantially improved performance.

*Table B.* Pilot results on challenging mathematics and coding tasks under W1.58A16 ternarization. CAT-Q+ denotes an enhanced variant of CAT-Q under the same quantization setting but with a new calibration data generation strategy.

| #Bits | Methods | MATH-500↑ | GSM8K↑ | Omni-MATH↑ | HumanEval+↑ | MBPP+↑ | Avg↑ |
|---|---|---|---|---|---|---|---|
| W16A16 | - | 96.80 | 88.10 | 34.64 | 85.37 | 64.81 | 73.94 |
| W1.58A16 | CAT-Q | 0.00 | 14.48 | 2.71 | 0.00 | 0.00 | 3.44 |
| W1.58A16 | CAT-Q+ | **58.40** | **61.56** | **14.93** | **53.98** | **39.15** | **45.60** |

## F. A More Comprehensive Comparison of Different Strategies for Determining $\alpha$ and $\triangle$.

This section visualizes the weight reconstruction errors under different strategies for determining the scaling factor $\alpha$ and the threshold $\triangle$ across multiple models and layers. Specifically, we show the results for the $4^{th}, 15^{th}$ and $30^{th}$ layers of Qwen3-4B (Figures C to E) and Llama2-7B (Figures F to H), as well as the MLP gate, up, and down projections of Qwen3-30B-A3B (Figures I to K), where each visualization includes multiple experts. Despite differences in model scale, architecture, and layer type, all figures exhibit highly consistent trends. Across all settings, static approximation (used in BitNet 1.58-bit families (Ma et al., 2024; Wang et al., 2025)) results in the largest reconstruction errors, indicating limited representational flexibility under fixed quantization statistics. Direct learning (initialized with the above static approximation used in BitNet 1.58-bit families) generally reduces the overall reconstruction error compared to static approximation. However, it also introduces noticeable outliers, with certain weight groups exhibiting errors even larger than those produced by static approximation. In contrast, the proposed learnable modulation consistently achieves the lowest reconstruction errors with significantly reduced variance, demonstrating both improved accuracy and stability. These observations consistently validate the effectiveness and robustness of our learnable modulation strategy across different models, layers, and expert configurations.

## G. Discussion of Limitations

Though we have provided two real-world 1.58-bit cases and shown their inference advantages to 2-bit and 4-bit formats, the current deployments remain under-optimized. BitNet's public 1.58-bit GPU and CPU kernels are limited to its own models. That is, optimized kernel support for diverse 1.58-bit LLMs remains lacking. Besides, although our CAT-Q is a universal, low-cost and accurate 1.58-bit PTQ method, there remains considerable room to reduce the performance gap to match the FP16 baseline, especially on complex reasoning tasks that are largely underexplored in LLM quantization research.

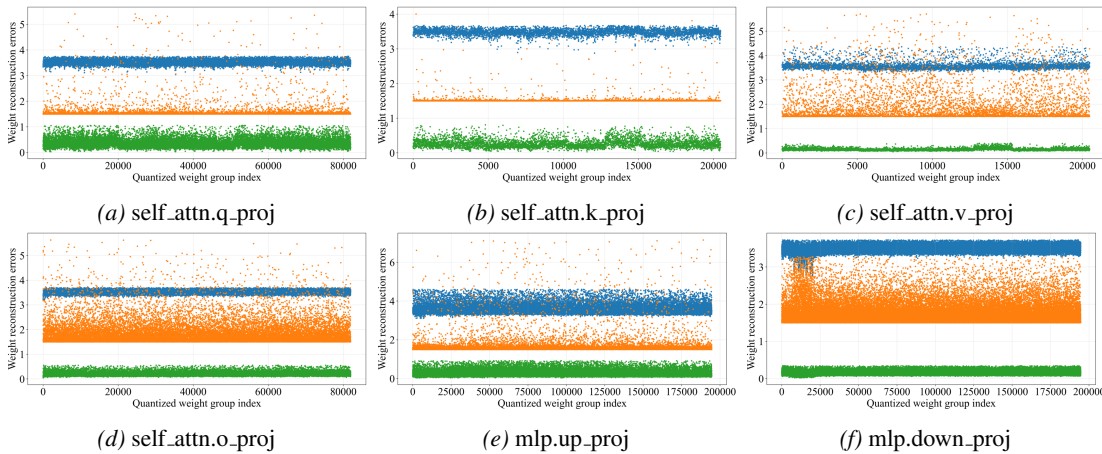

*(a)* self_attn.q_proj        *(b)* self_attn.k_proj        *(c)* self_attn.v_proj

*(d)* self_attn.o_proj        *(e)* mlp.up_proj        *(f)* mlp.down_proj

*Figure C.* Comparison of weight reconstruction errors with the scaling factor $\alpha$ and the threshold $\Delta$ determined by static approximation (blue dots), direct learning (orange dots) and our learnable modulation (green dots). Under the same ternarization setting, we use the $4^{th}$ layer of Qwen3-4B for an illustration.

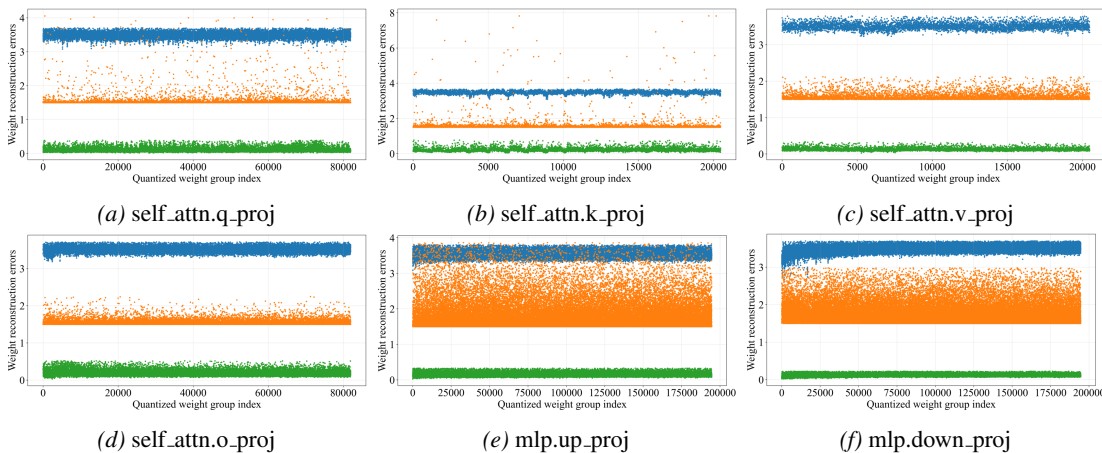

*(a)* self_attn.q_proj        *(b)* self_attn.k_proj        *(c)* self_attn.v_proj

*(d)* self_attn.o_proj        *(e)* mlp.up_proj        *(f)* mlp.down_proj

*Figure D.* Comparison of weight reconstruction errors with the scaling factor $\alpha$ and the threshold $\Delta$ determined by static approximation (blue dots), direct learning (orange dots) and our learnable modulation (green dots). Under the same ternarization setting, we use the $15^{th}$ layer of Qwen3-4B for an illustration.

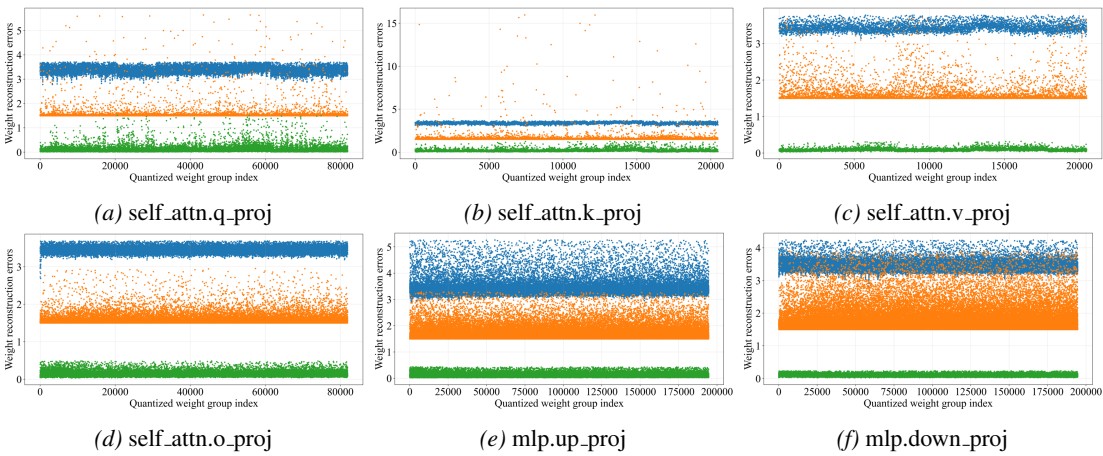

*(a)* self_attn.q_proj        *(b)* self_attn.k_proj        *(c)* self_attn.v_proj

*(d)* self_attn.o_proj        *(e)* mlp.up_proj        *(f)* mlp.down_proj

*Figure E.* Comparison of weight reconstruction errors with the scaling factor $\alpha$ and the threshold $\Delta$ determined by static approximation (blue dots), direct learning (orange dots) and our learnable modulation (green dots). Under the same ternarization setting, we use the $30^{th}$ layer of Qwen3-4B for an illustration.

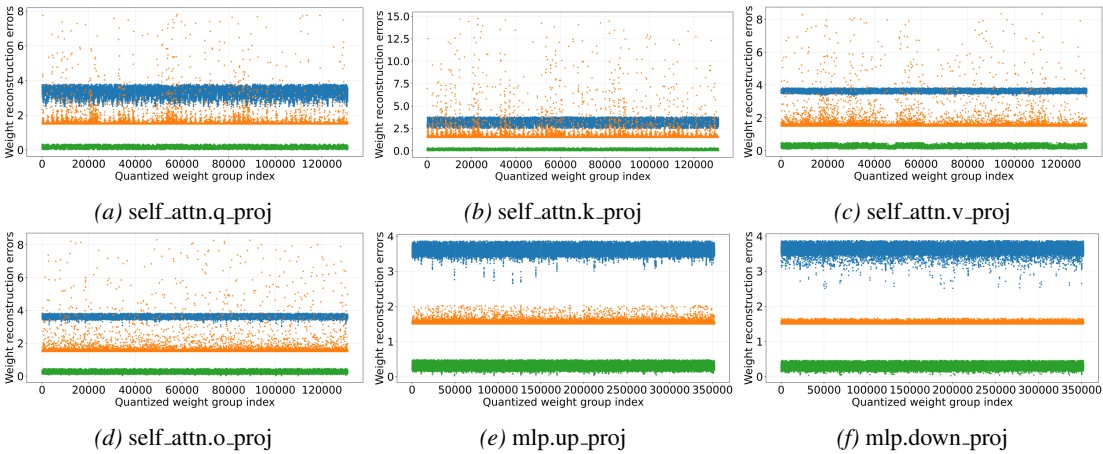

*Figure F.* Comparison of weight reconstruction errors with the scaling factor $\alpha$ and the threshold $\Delta$ determined by static approximation (blue dots), direct learning (orange dots) and our learnable modulation (green dots). Under the same ternarization setting, we use the $4^{th}$ layer of Llama2-7B for an illustration.

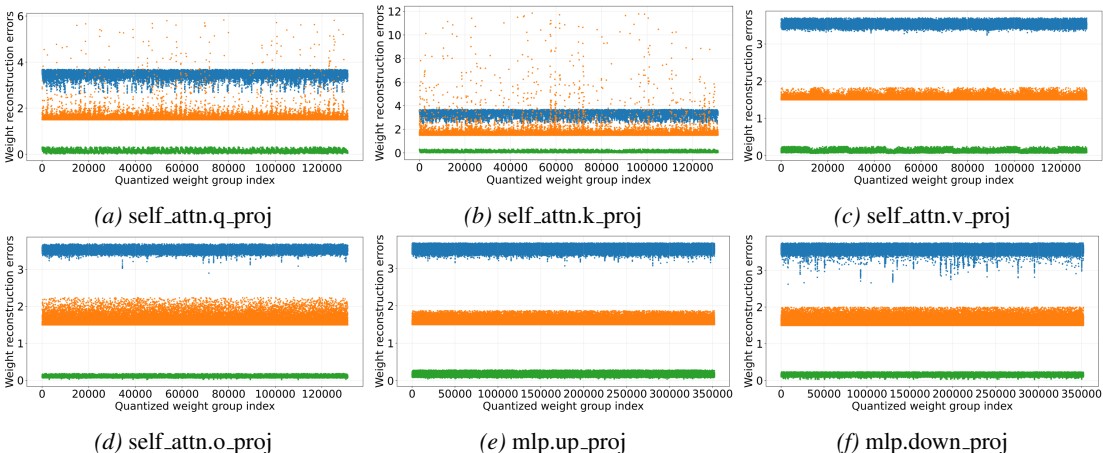

*Figure G.* Comparison of weight reconstruction errors with the scaling factor $\alpha$ and the threshold $\Delta$ determined by static approximation (blue dots), direct learning (orange dots) and our learnable modulation (green dots). Under the same ternarization setting, we use the $15^{th}$ layer of Llama2-7B for an illustration.

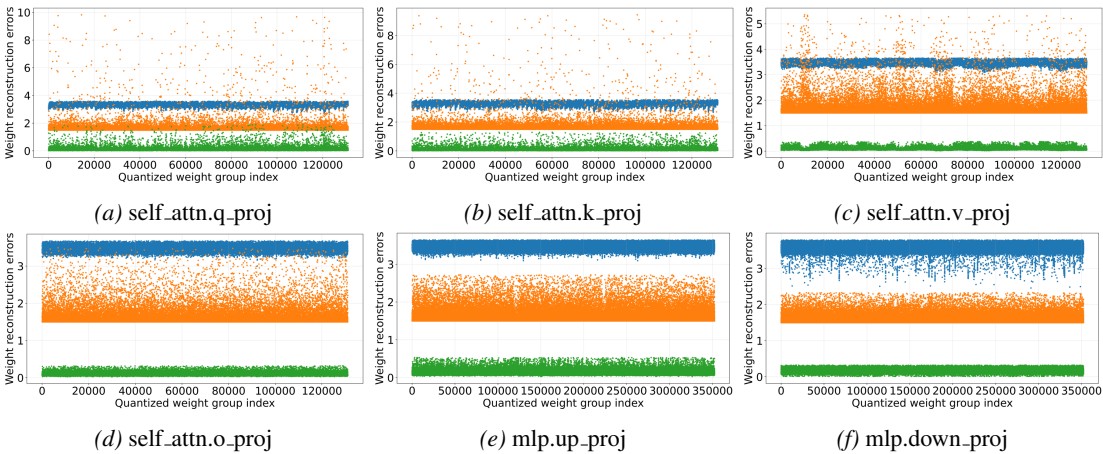

*Figure H.* Comparison of weight reconstruction errors with the scaling factor $\alpha$ and the threshold $\Delta$ determined by static approximation (blue dots), direct learning (orange dots) and our learnable modulation (green dots). Under the same ternarization setting, we use the $30^{th}$ layer of Llama2-7B for an illustration.

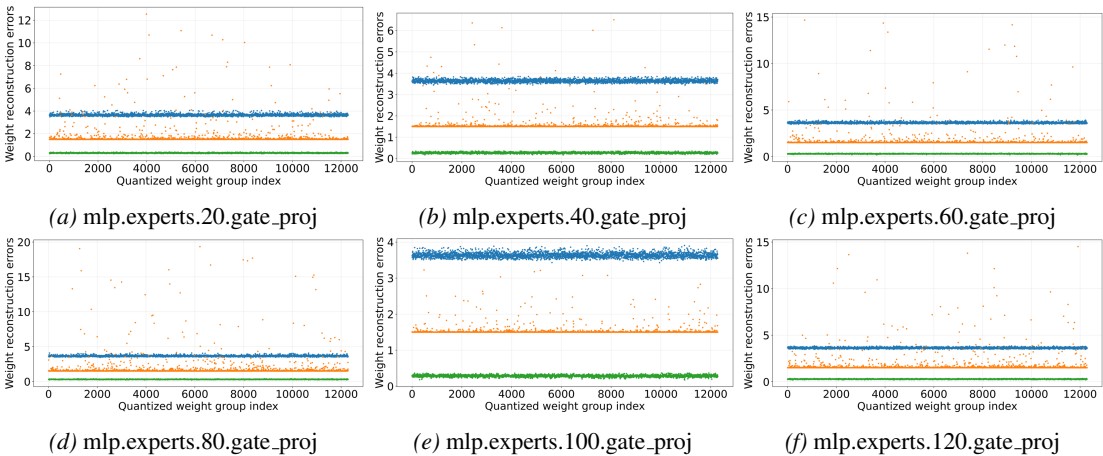

*(a)* mlp.experts.20.gate_proj     *(b)* mlp.experts.40.gate_proj     *(c)* mlp.experts.60.gate_proj

*(d)* mlp.experts.80.gate_proj     *(e)* mlp.experts.100.gate_proj     *(f)* mlp.experts.120.gate_proj

*Figure I.* Comparison of weight reconstruction errors with the scaling factor $\alpha$ and the threshold $\Delta$ determined by static approximation (blue dots), direct learning (orange dots) and our learnable modulation (green dots). Under the same ternarization setting, we use the $15^{th}$ layer (**gate_proj**) of Qwen3-30B-A3B, where six experts are uniformly sampled from the total 128 experts for an illustration.

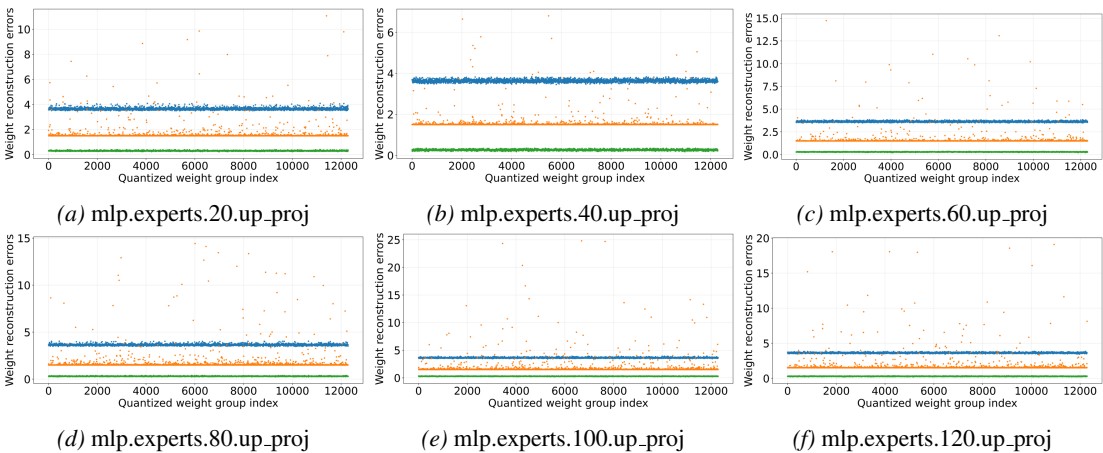

*(a)* mlp.experts.20.up_proj     *(b)* mlp.experts.40.up_proj     *(c)* mlp.experts.60.up_proj

*(d)* mlp.experts.80.up_proj     *(e)* mlp.experts.100.up_proj     *(f)* mlp.experts.120.up_proj

*Figure J.* Comparison of weight reconstruction errors with the scaling factor $\alpha$ and the threshold $\Delta$ determined by static approximation (blue dots), direct learning (orange dots) and our learnable modulation (green dots). Under the same ternarization setting, we use the $15^{th}$ layer (**up_proj**) of Qwen3-30B-A3B, where six experts are uniformly sampled from the total 128 experts for an illustration.

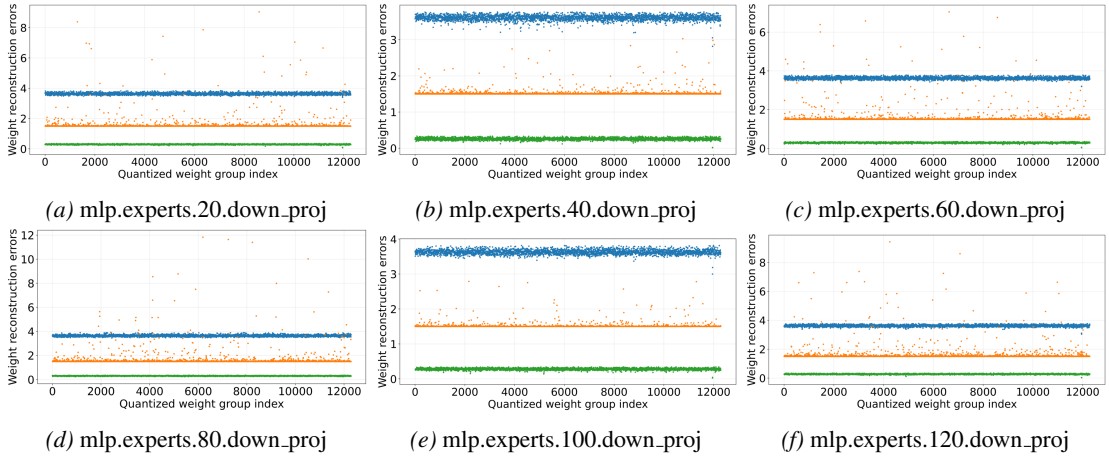

*(a)* mlp.experts.20.down_proj     *(b)* mlp.experts.40.down_proj     *(c)* mlp.experts.60.down_proj

*(d)* mlp.experts.80.down_proj     *(e)* mlp.experts.100.down_proj     *(f)* mlp.experts.120.down_proj

*Figure K.* Comparison of weight reconstruction errors with the scaling factor $\alpha$ and the threshold $\Delta$ determined by static approximation (blue dots), direct learning (orange dots) and our learnable modulation (green dots). Under the same ternarization setting, we use the $15^{th}$ layer (**down_proj**) of Qwen3-30B-A3B, where six experts are uniformly sampled from the total 128 experts for an illustration.

