# OpenReview forum: "CAT-Q: Cost-efficient and Accurate Ternary Quantization for LLMs"
_ICML.cc/2026/Conference — ICML 2026 spotlight_

### Official Review · Reviewer_LXB9 · 2026-03-07

**Soundness:** 3
**Presentation:** 3
**Significance:** 2
**Originality:** 2
**Overall Recommendation:** 5
**Confidence:** 4

**Summary:**

This paper presents CAT-Q, which is a post-training framework for ternarizing LLMs. The authors mentioned two challenges in ternary weight quantization in PTQ: (i) distribution misalignment of ternary weight,  (ii) difficulty of stable convergence caused by extreme discrete points.

Authors propose two method, one is learnable modulation, which uses three learnable factors for ternary quantization, the separation of scale \alpha and ternary threshold \Delta makes the optimization more flexible. The second method is softened ternarization, which transforms the hard quantization into soft and differentiable one using tanh functions. Experiments on diverse pre-trained LLMs shows the effectiveness of CAT-Q and its generalizability.

**Compliance With Llm Reviewing Policy:**

Affirmed.

**Final Justification:**

The rebuttal has effectively addressed my main concerns. And the additional explanations and comprehensive experiments have strengthened the paper.  I hope the experimental results during the rebuttal period, detailed settings as well as the codes for finetuning can be included in the final version of the paper. I believe the method is valuable in ultra-low bit ternary quantization for LLMs.
See my last comment for more details.

**Key Questions For Authors:**

1) The performance on more complex tasks mentioned in Weaknesses.

2) The contribution of sliding-layer reconstruction compared to layerwise or block-wise reconstruction.

3) Comparison of calibration/reconstruction time of existing PTQ methods.

**Limitations:**

Not discussed.

Their methods are evaluated on accuracy benchmarks and do not include real deployment latency tests.
The experiments on common sense QA tasks narrow in task scope, so the robustness of the proposed method is unclear. The paper seems to have no negative social impact.

**Strengths And Weaknesses:**

Strengths:

1) This paper is technically sound. It solves the problem of large-scale retraining of extremely low bitwidth quantization. It uses 8 A100 GPUs and trains for 60 hours, which is relatively efficient compared to existing QAT methods.

2) The model sizes in the experiments are diverse, ranging from 1.7B to 235B, which forcefully shows the practical generalization of the proposed method across model size and model family.

3) The experimental analysis is abundant and comprehensive, which showcases the contribution of LM and ST.

4) The representation is clear and easy to follow. The organization of the content is good.

5) The problem is not new, but the performance of the proposed method seems to be good. It saves much training time compared to QAT methods, and the performance is slightly better than other PTQ binarization methods.



Weaknesses:

1) The novelty contribution is somewhat incremental. The proposed learnable modulation is already applied in previous works, such as [1]. Soft quantization using tanh functions is also not new [2]. Sliding-layer optimization is brought from SliderQuant. Therefore, I think the novelty of this paper is not strong. The authors combine previous best practices and build a practical implementation of ternary quantization on LLMs, but not a new quantization paradigm.

2) The ablation studies are only conducted on the LM and ST, but not on the sliding-layer reconstruction. It is necessary to decouple the contribution of sliding-layer with layer-wise or block-wise reconstruction.

3) Although the authors claim that the reconstruction time of their method is short, with 8*A100 GPUs for 60 hours, it is unclear whether this is faster than other PTQ methods. It lacks experimental evidence about the calibration time between existing PTQ methods on LLMs.

4) The authors only tested their method on common sense QA tasks, which is a little bit easy especially for larger models with 7B or more parameters. And the performance drop is still very significant even if it performs better than existing methods. I wonder how the performance on more complex tasks, such as longer context, multi-modal, or reasoning.


Reference:
[1] Vindas Y, Roux E, Guépié BK, Almar M, Delachartre P. An asymmetric heuristic for trained ternary quantization based on the statistics of the weights: An application to medical signal classification. Pattern Recognition Letters. 2024.

[2] Gong R, Liu X, Jiang S, Li TH, Hu P, Lin J, et al. Differentiable Soft Quantization: Bridging Full-Precision and Low-Bit Neural Networks. IEEE International Conference on Computer Vision. 2019.

---

> ### Author Rebuttal · Authors · 2026-03-31
>
> Thanks for your constructive comments.
>
> **1. Responses to Weakness 1**: We argue that our work's novelty is not "somewhat incremental": 1) [1] uses TTQ(Zhu et al., ICLR2017) that learns two scaling factors to approximate positive & negative weights. Our LM learns a single scaling factor by the disentangled weights redistribution and approximation. DSQ(Gong et al., ICCV2019) [2] is an integer-bit QAT method based on a standard tanh function, which cannot handle 1.58-bit quantization. Our ST proposes a relay of differentiable ternarization and hard ternarization by a novel combination of tanh functions; 2) TTQ and [1-2] focus on quantizing CNNs and require fine-tuning, while our CAT-Q is a fast PTQ method for LLMs; 3) In the Method section, we had discussions of CAT-Q with related works including TTQ & [2] (Line139-153) and SliderQuant (Line230-244), and comparisons in Table 7/8/5.
>
> **2. To Key Question 1(Weakness 4)** about the performance on more complex tasks.
>
> **Responses**: As existing 1.58-bit methods (e.g. BitNet V1/V2, TriLM, Tequila) mainly consider commonsense QA tasks, we follow them in benchmarking. Recently, SliderQuant reported its and OmniQuant’s 2-bit results on complex math and code tasks, whose sliding-layer reconstruction is used as our base framework. In Table A below, we compare CAT-Q with them on Qwen-14B distilled by DeepSeek-R1, and with SliderQuant* (its 1.58-bit implementation by us) on Qwen3-4B/-14B, validating the superiority of our CAT-Q.
>
> **Table A**: Comparison on 5 challenging math and code tasks.
> |Model|#Bits|Method|Math-500↑|AIME-2024↑|GSM8K↑|HumanEval+↑|MBPP+↑|Avg↑|
> |-|:-:|:-:|:-:|:-:|:-:|:-:|:-:|:-:|
> |Qwen-14B|W16A16|-|95.00|73.33|91.50|73.17|61.11|78.82|
> |Qwen-14B|W2A16|OmniQuant|0.00|0.00|2.20|0.00|0.00|0.44|
> |Qwen-14B|W2A16|SliderQuant|29.40|10.00|54.28|12.80|21.16|25.53|
> |Qwen-14B|W1.58A16|SliderQuant*(1M)|8.40|3.33|24.34|3.66|5.82|9.11|
> |Qwen-14B|W1.58A16|CAT-Q(1M)|42.60|16.67|60.65|36.00|33.86|37.96|
> |Qwen-14B|W1.58A16|CAT-Q(2M)|**53.20**|**20.00**|**75.82**|**45.12**|**42.33**|**47.29**|
> |Qwen3-4B|W16A16|-|96.80|73.33|88.10|85.37|64.91|81.70|
> |Qwen3-4B|W1.58A16|SliderQuant*(1M)|3.20|0.00|8.34|2.44|3.70|3.54|
> |Qwen3-4B|W1.58A16|CAT-Q(1M)|29.20|13.30|43.00|28.04|23.01|27.31|
> |Qwen3-4B|W1.58A16|CAT-Q(2M)|**36.40**|**16.67**|**53.75**|**35.37**|**28.84**|**34.21**|
>
> **3. To Key Question 2(Weakness 2)** about the contribution of sliding-layer compared to layer-wise/block-wise reconstruction.
>
> **Responses**: Table 5 of our paper shows an ablation on Qwen3-4B and Llama2-7B to study the gains of our LM and ST modules to sliding-layer reconstruction SliderQuant (the first row denotes its results). In Table B below, we further compare SliderQuant with layer-wise/block-wise reconstruction and LM/ST, showing that the major gains are attributed to our LM and ST modules.
>
> **Table B**: The role of sliding-layer reconstruction.
> |Model|Method|PIQA↑|ARC-e↑|ARC-c↑|HellaSwag↑|Winogrande↑|Avg↑|
> |-|:-:|:-:|:-:|:-:|:-:|:-:|:-:|
> |Qwen3-4B|Layer-wise|52.10|27.80|24.12|33.78|46.65|36.89|
> |Qwen3-4B|Block-wise|53.93|29.58|25.33|35.19|48.01|38.41|
> |Qwen3-4B|SliderQuant|55.33|31.99|26.11|36.70|50.67|40.16|
> |Qwen3-4B|SliderQuant+LM|68.66|56.95|33.87|48.30|58.19|53.19|
> |Qwen3-4B|SliderQuant+LM+ST(CAT-Q)|**70.62**|**62.29**|**36.95**|**53.24**|**62.19**|**57.06**|
> |Llama2-7B|Layer-wise|59.60|38.20|25.83|40.50|47.45|42.32|
> |Llama2-7B|Block-wise|61.20|39.00|25.90|41.60|48.15|43.17|
> |Llama2-7B|SliderQuant|63.06|41.29|27.05|43.53|51.93|45.37|
> |Llama2-7B|SliderQuant+LM|72.69|58.98|31.55|57.85|58.77|55.97|
> |Llama2-7B|SliderQuant+LM+ST(CAT-Q)|**72.91**|**60.06**|**33.62**|**60.95**|**60.93**|**57.69**|
>
> **4. To Key Question 3 (Weakness 3)** about the comparison of calibration time with existing PTQ methods.
>
> **Responses**: Your mentioned "with 8\*A100 GPUs for 60 hours" is for the 235B model. Generally, our CAT-Q needs 1 to 60 hours (with 8\*A100 GPUs) to quantize 1.7B to 235B LLMs, which are multiple orders of magnitude faster than existing 1.58-bit QAT methods. Table C below shows a comparison of calibration time of CAT-Q with SliderQuant, and other PTQ methods tested in EfficientQAT (a two-stage PTQ method quantizes pre-trained LLMs with re-training), showing CAT-Q's competitiveness in efficiency & accuracy.
>
> **Table C**: Calibration time comparison with 2-bit PTQ methods on Llama2-70B. `n1|n2`in`#Samples` denotes the method has two stages. * denotes our reproduced results using the public code. Avg is over 5 commonsense QA tasks.
> |Method|#Samples|Calibration Time(A100 hours)|#Bits|Avg↑|
> |-|:-:|:-:|:-:|:-:|
> |AQLM|4096|336|W2A16|69.85|
> |EfficientQAT|4096\|4096|41|W2A16|68.93|
> |EfficientQAT*|8192\|8192|278|W2A16|70.35|
> |QUIP#|6144\|256|300|W2A16|70.91|
> |DB-LLM|20K|82|W1(dual 1-bit)A16|65.82|
> |SliderQuant|128|64|W2A16|71.08|
> |SliderQuant*|512|256|W1.58A16|56.37|
> |CAT-Q|512|256|W1.58A16|**72.72**|
>
> **5.** GPU/CPU latency tests and limitation discussion are in our responses to Reviewer eGPr's concern 2/4.

---

> > ### Author Rebuttal · Reviewer_LXB9 · 2026-04-01
> >
> > I have read the authors’ responses and I appreciate the additional explanations and evidences. The rebuttal is sufficient to address my concerns on the other questions. However, I remain unconvinced about the novelty of the paper.
> >
> > 1) The authors clarifies that the proposed method is not identical to prior work, but I think it cannot support the novelty of the paper. The contribution of this paper is to combine and verify the effectiveness to the ternary PTQ. I appreciate the authors' efforts to adapt the existing formulations to the extreme-low bit LLMs. But the modules and mechanisms are not the first time to be proposed.
> >
> > 2) The ablation study of sliding-layer reconstruction is convincing, which solves my concern. And the experimental results for complex tasks also answer my concern. Thank the authors for the honest and quick evaluation on these additional benchmarks.
> >
> > 3) The calibration time of the proposed method is too long for it to be considered a PTQ method. Usually, PTQ methods are expected to be fast and lightweight, which may spend several to tens of GPU-hours for a 7B to 70B LLM. If the time is excessively long, it may lose the practical advantage of PTQ and could be resembled as a QAT method for further improvement. Especially in ultra low-bit setting, if it requires such a long calibration time, it is not limited to be a PTQ method. But it is not a problem for this paper specifically, since the other ternary/binary PTQ methods may also have similar calibration time.

---

> > > ### Author Response · Authors · 2026-04-03
> > >
> > > Thank you for the detailed feedback. We are glad to see that our rebuttal is mostly recognized by you.
> > >
> > > **1. To your remaining concern** about the novelty of our paper, **in greater detail, our paper differs from prior works in two parts**:
> > >
> > > **Part A: focus and quantization paradigm**: **1)** Your referred [1] tackles medical image classification by 1.58-bit conv. neural networks (CNNs) based on TTQ [Zhu et al., ICLR 2017] cited in our paper; **2)** Your referred [2], namely DSQ [Gong et al., ICCV 2019] cited in our paper, addresses uniform integer-bit (e.g., 4-bit/2-bit) but not 1.58-bit quantization; **3)** On one side, [1], TTQ and DSQ[2] focus on quantizing small CNNs for computer vision tasks. On the other side, they rely on QAT, requiring re-training/fine-tuning on task data; **4)** Unlike them, our paper aims to bridge a critical gap in 1.58-bit LLM research by a cost-efficient, scalable and accurate PTQ method using task-agnostic data: existing SOTA 1.58-bit methods rely on data-intensive and costly QAT (10B to 300B  training tokens), which are limited to specific small-sized architectures (at most 7B model size).
> > >
> > > **Part B: method formulation**: The concept of 1.58-bit quantization is proposed in TWN [Li et al., arXiv 2016], approximating the high-precision weights $W$ of a layer by$$W\approx\begin{cases}\alpha,&\text{if }W>\Delta\\\\ 0,&\text{if }W\in[-\Delta,\Delta]\\\\-\alpha,&\text{if }W<-\Delta\end{cases}\tag{1}$$Technically, how to compute 1) the scaling factor $\alpha$, 2) the threshold $\Delta$, and 3) gradients (ternarization is not differentiable), are three key problems. BitNet, TriLM, Tequila and our CAT-Q use this simple 1.58-bit format, but [1] uses TTQ where$$W\approx\begin{cases}\alpha_p,&\text{if }W>\Delta_{max}\\\\ 0,&\text{if }W\in [\Delta_{min},\Delta_{max}]\\\\-\alpha_n,&\text{if }W<\Delta_{min}\end{cases}\tag{2}$$That is, [1] and TTQ use two sets of scaling factor and threshold to asymmetrically approximate positive and negative weights of $W$, incurring extra overhead at inference. In formulation, [1] and TTQ 1) learn $\alpha_p,\alpha_n$ based on $W$, but 2) compute $\Delta_{max},\Delta_{min}$ as fixed statistics, and 3) use hard ternarization by straight-through estimator, following TWN. In contrast, LM, our first core module of CAT-Q, has two novel designs via three learnable factors $\delta_\mu,\delta_\alpha,\delta_\Delta$: 1) an adaptive linear transformation (Eq.3 of our paper) using $\delta_\mu,\delta_\alpha$ to redistribute $W$ into $\hat{W}$ that is easier to ternarize; 2) a disentangled optimization scheme that uses $\hat{W}$ but not $W$ to learn $\delta_\mu,\delta_\alpha,\delta_\Delta$ (estimate $\alpha, \Delta$) and the simple 1.58-bit weights approximation (Eq.1 above), promoting ternarization accuracy while preserving the hardware-friendly inference property of TWN, as shown in Table 8/7/9. Our second core module ST further addresses the non-differentiable issue by two novel designs coupling LM: 1) a transition function (Eq.5 of our paper) using a pair of symmetric tanh functions $f(\hat{W};s,\Delta)=\frac{\tanh(s(\hat{W}-\Delta))+\tanh(s(\hat{W}+\Delta))}{2\tanh(s)}$, enabling differentiable ternarization; 2) a two-stage relay of differentiable to hard ternarization, ensuring stable convergence in PTQ manner. Similar to LSQ [Esser et al., ICLR 2020] cited in our paper, DSQ [2] uses a standard tanh function $k\tanh(s(W-m_i))$ without weights redistribution to learn step sizes for uniform integer-bit (e.g., 4-bit/2-bit) quantization of small CNNs by QAT. We use sliding-layer reconstruction PTQ framework SliderQuant when implementing our CAT-Q. Therefore, the learning-based concept of scaling factor or tanh function is not new in quantization, but our CAT-Q differs from prior works with multiple novel contributions clarified above.
> > >
> > > **In Line 55-86(left column), 39-152(left), 157-164(left), 139-153(right), 230-244(left) of our paper**, we properly discussed our method's connections and differences to prior works. We will improve them further.
> > >
> > > **2. Extra clarification**: We appreciate that you think "The calibration time...is not a problem for this paper...". For LLMs, 1.58-bit quantization is much harder to converge than conventional quantization (e.g., 8-bit/4-bit), due to its extreme low-bit format {1,0,-1}. Yes, our CAT-Q is not as fast as conventional PTQ methods, yet it is a PTQ method as it uses task-agnostic samples but not training/task data for calibration. Aligned with the above research gap in 1.58-bit LLMs, our CAT-Q only needs 1M tokens with 1 to 60 hours on a 8×A100 GPU server to ternarize pre-trained 1.7B to 235B LLMs (dense & MoE), while existing SOTA 1.58-bit LLMs (BitNet, TriLM, Tequila) having at most 7B model size need 30B to 300B training tokens and large-scale GPU cluster (e.g., 256×A100). Our work makes scalable LLM ternarization technology accessible to the vast majority of researchers for the first time.
> > >
> > > Looking forward to your feedback!

---

### Official Review · Reviewer_eGPr · 2026-03-09

**Soundness:** 2
**Presentation:** 3
**Significance:** 2
**Originality:** 3
**Overall Recommendation:** 4
**Confidence:** 4

**Summary:**

The paper proposes CAT-Q, a Post-Training Quantization method designed to compress Large Language Models into ternary weights (1.58-bit). To address the severe performance degradation typical of extreme quantization without relying on costly Quantization-Aware Training (QAT), the authors introduce two main components: Learnable Modulation and Softened Ternarization . LM uses learnable factors to adapt the pre-trained weight distributions and ternary thresholds, making them less sensitive to quantization. ST employs a continuous transition function to smoothly guide the optimization from differentiable ternarization to hard ternarization. The authors demonstrate that CAT-Q can quantize models ranging from 1.7B to 235B parameters using only 512 calibration samples, achieving competitive zero-shot reasoning performance compared to resource-intensive QAT baselines.

**Compliance With Llm Reviewing Policy:**

Affirmed.

**Final Justification:**

**Final Justification**

I recommend **Weak Accept**.

**Summary:**

**Strengths:** The proposed CAT-Q method presents a compelling solution to extreme low-bit quantization by combining Layer-wise Minimization (LM) and Sliding-layer Ternary (ST) within a reconstruction pipeline. The approach effectively addresses the instability of 1.58-bit PTQ, and its cost-efficiency—requiring only 512 calibration samples compared to 100B+ tokens for QAT methods—significantly lowers the barrier to deploying ternary LLMs.

**Weaknesses:** My initial concerns centered on three gaps: (1) limited benchmark coverage (only five commonsense tasks, lacking perplexity, mathematical reasoning, and long-context evaluation); (2) absence of hardware profiling and practical deployment metrics; and (3) failure to demonstrate tangible advantages against optimized real-world baselines like llama.cpp's 4-bit implementations.

**Rebuttal Impact:** The authors' rebuttal addressed my primary concerns regarding validation. Specifically, the addition of actual speedup comparisons against highly optimized deployment solutions (e.g., llama.cpp variants) directly tackles the critical gap between theoretical bit-width reduction and practical performance gains.

**Key Questions For Authors:**

**1.Comprehensive Task Evaluation:** Could you provide perplexity scores (e.g., on WikiText2 ) and evaluations on generative tasks (e.g., MMLU-Pro, GSM8K， LongBench) to prove CAT-Q's stability beyond multiple-choice zero-shot reasoning?

**2. Hardware Profiling:** Can you provide actual memory consumption (peak memory during generation), end-to-end latency, and token throughput measurements of your quantized models on real hardware (e.g., A100, RTX 3090/4090, or edge environments)?

**3. Practical Baselines:** How does CAT-Q compare to widely adopted deployment formats? Please include a comparison against baselines like llama.cpp (and its relevant quantization schemes) or QUIP#, specifically mapping the trade-off between actual inference throughput and accuracy. I strongly consider raising my score if it can be shown that CAT-Q achieves superior accuracy under similar physical memory/speed constraints.

**Limitations:**

No. Authors should explicitly discuss the current lack of hardware support/kernels for their specific format, as well as any potential performance/throughput degradation that occurs at 1.58 bits.

**Strengths And Weaknesses:**

**Strengths:**
- **Methodology:** The combination of LM and ST within a sliding-layer output reconstruction pipeline is an effective approach to the notoriously unstable process of PTQ for extreme low-bit quantization.
- **Cost-Efficiency:** By requiring only 512 calibration samples (roughly 1 million tokens), CAT-Q drastically reduces the compute bottleneck associated with 1.58-bit QAT methods (which often require 100B+ tokens), making ternary LLMs highly accessible.

**Weaknesses:**
- **Limited Evaluation Benchmarks:** The evaluation is overly narrow. The work only evaluates CAT-Q on five zero-shot commonsense reasoning benchmarks (PIQA, ARC, HellaSwag, Winogrande). It critically lacks evaluation on other standard LLM capabilities such as language modeling (perplexity), mathematical reasoning (MMLU), and long-context understanding.
- **Lack of Hardware Profiling & Practical Baselines:** As a quantization method aimed at LLM compression and acceleration, relying purely on theoretical bit-width and zero-shot accuracy is insufficient. The paper completely omits specific experimental data on inference speedup ratios, memory consumption reduction, and actual throughput on target hardware (e.g., A100, RTX 3090, or edge devices).
- **Real-world Deployment Gap:** The current landscape of LLM quantization is saturated. Practitioners do not adopt a model simply because the theoretical footprint is 0.5 bits lower; they adopt methods that translate to tangible latency and throughput improvements. Real-world deployments heavily rely on highly optimized, albeit unpublished, solutions like **4-bit model** from llama.cpp (e.g., ik_llama.cpp quants) or structured methods like QUIP#. Demonstrating that CAT-Q can achieve higher accuracy at similar actual memory/speed limits—or faster speeds at similar accuracy limits—against these deployment-ready baselines is crucial to proving its superiority.

---

> ### Author Rebuttal · Authors · 2026-03-31
>
> Thanks for your constructive comments.
>
> **1. To your Key Question 1(Weakness 1)** about more comprehensive task evaluation.
>
> **Our responses**: In 1.58-bit LLM quantization research, existing SOTA methods such as Tequila (ICLR 2026) and TriLM (ICLR 2025)  mainly consider commonsense reasoning tasks, so we followed them in benchmarking. To have a more comprehensive evaluation, **1) in Table A below**, we test our CAT-Q with Llama2-7B|Qwen3-4B on language modeling (WikiText2) and multi-task QA (MMLU), and compare with 2-bit results reported in recent PTQ works (SliderQuant, QUIP#), and 1.58-bit methods. We also compare CAT-Q with SliderQuant* (its 1.58-bit implementation by us) as its sliding layer reconstruction is used as our base framework. We can see: for Llama2-7B, our 1.58-bit model gets 8.51 perplexity on WikiText2 which is significantly better than SliderQuant*, and is close to 2-bit model of QUIP#. On MMLU with Llama2-7B|Qwen3-4B, our method is superior to SliderQuant* and other methods; **2) in our responses to Reviewer LXB9's Key Question 1(Weakness 4)**, we add a comparison on 5 challenging math and code tasks including GSM8K, Math-500, AIME-2024, HumanEval+ and MBPP+ with long context, further showing the superiority of our CAT-Q.
>
> **Table A**: Comparison on WikiText2 (perplexity) and MMLU (accuracy). `NA` denotes the method does not report results for this task.
>  |Model|#Bits|Method|Method Type|WikiText2↓|MMLU↑|
> |-|:-:|:-:|:-:|:-:|:-:|
> |Llama2-7B|W16A16|-|-|5.47|40.85|
> |Llama2-7B|W2A16|OmniQuant|PTQ|37.37|19.78|
> |Llama2-7B|W2A16|QUIP#|PTQ|**6.66**|NA|
> |Llama2-7B|W2A16|SliderQuant|PTQ|9.59|24.26|
> |Llama2-7B|W1.58A16|SliderQuant*|PTQ|26.22|21.34|
> |BitNetV2-7B|W1.58A16|BitNetV2|QAT|NA|NA|
> |Llama2-7B|W1.58A16|CAT-Q|PTQ|8.51|**27.32**|
> |Qwen3-4B|W16A16|-|-|13.66|68.27|
> |Qwen3-4B|W1.58A16|SliderQuant*|PTQ|45.14|33.58|
> |TriLM-3.9B|W1.58A16|TriLM|QAT|NA|32.80|
> |TequilaLLM-3B|W1.58A16|TequilaLLM|QAT|NA|NA|
> |Qwen3-4B|W1.58A16|CAT-Q|PTQ|**18.99**|**44.03**|
>
> **2. To your Key Question 2(Weakness 2)** about hardware profiling with our 1.58-bit models.
>
> **Our responses**: We now have your requested hardware profiling tests on hand. Concretely, we have deployed our 1.58-bit Qwen3-4B and Llama2-7B using `llama.cpp` with its ternary implementation (TQ2_0 for GPU, TQ1_0 for CPU), and compare inference with 4-bit (Q4_K_M) and 2-bit (Q2_K_M) baselines on GPU and CPU. Table B and C below shows detailed results for language modeling with batch size 1 and generation length 512. Compared to the 4-bit baseline, our 1.58-bit Qwen3-4B|Llama2-7B reduces weight memory by 49.57%|50.00% on GPU and 56.47%|57.37% on CPU, while yielding 1.42x|1.62x GPU and 1.93x|2.22x CPU decoding speedup, respectively. Notably, compared to the 2-bit baseline, our 1.58-bit Qwen3-4B|Llama2-7B reduces weight memory by 24.52%|27.76% on GPU and 34.84%|38.40% on CPU, while yielding 1.19x|1.34x GPU and 1.57x|1.55x CPU decoding speedup, respectively.
>
> **Table B**: Model profiling with `llama.cpp` on an RTX 4090 GPU.
> |Model|Deployment Format|Weight Memory (GB)|Decoding Throughput (tokens per sec)|
> |-|:-:|:-:|:-:|
> |Qwen3-4B|Q4_K_M (4-bit)|2.32|225.91|
> |Qwen3-4B|Q2_K_M (2-bit)|1.55|268.51|
> |Qwen3-4B|TQ2_0 (ternary)|**1.17**|**320.34**|
> |Llama2-7B|Q4_K_M (4-bit)|3.80|174.34|
> |Llama2-7B|Q2_K_M (2-bit)|2.63|210.72|
> |Llama2-7B|TQ2_0 (ternary)|**1.90**|**281.66**|
>
> **Table C**: Model profiling with `llama.cpp` on Intel Core i9-13900KF CPU.
> |Model|Deployment Format|Weight Memory|Decoding Throughput (tokens per sec)|
> |-|:-:|:-:|:-:|
> |Qwen3-4B|Q4_K_M (4-bit)|2.32|19.31|
> |Qwen3-4B|Q2_K_M (2-bit)|1.55|23.84|
> |Qwen3-4B|TQ1_0 (ternary)|**1.01**|**37.35**|
> |Llama2-7B|Q4_K_M (4-bit)|3.80|11.83|
> |Llama2-7B|Q2_K_M (2-bit)|2.63|17.01|
> |Llama2-7B|TQ1_0 (ternary)|**1.62**|**26.30**|
>
> **3. To your Key Question 3(Weakness 3)** about comparison with real-world deployment baseline formats.
>
> **Our responses**: According to the results in the above Table A/B/C and in our paper's Table 2/3/4, we can clearly show: in terms of accuracy and inference memory/speed, our 1.58-bit models (e.g., Llama2-7B|-70B) are superior to existing 2-bit models including GPTQ/AWQ supported in `llama.cpp`, SliderQuant and others. Indeed, 4-bit format having better accuracy is widely used, but 1.58-bit format is attracting increased attention due to better efficiency.
>
> **4 A discussion on limitations:** Though we have provided two real 1.58-bit cases and shown their inference advantages to 2-bit & 4-bit formats, the current deployments remain under-optimized. BitNet's public 1.58-bit GPU/CPU kernels are limited to its own models. So, optimized kernel support for diverse 1.58-bit LLMs remains lacking. We have an ongoing project to address this gap. Besides, our CAT-Q is a universal, low-cost and accurate 1.58-bit PTQ method, there remains large room to reduce the performance gap to match the FP16 baseline, especially on complex reasoning tasks that are underexplored in LLM quantization research.

---

> > ### Author Rebuttal · Reviewer_eGPr · 2026-04-03
> >
> > Thank you for your efforts. The evaluation on other tasks and the hardware profiling are convincing.
> > Regarding the llama.cpp comparison, I would like to see a fair comparison using IQ1_Quant (specifically IQ1_S and IQ1_M) in terms of PPL and speed. I hope you can supplement this experiment in your subsequent response.
> > Overall, I believe the outcome will be positive, given CAT-Q's dedicated design and the PTQ efforts. Therefore, I will raise my score.

---

> > > ### Author Response · Authors · 2026-04-04
> > >
> > > We greatly appreciate your insightful feedback and are glad that our rebuttal has addressed your concerns.
> > >
> > > Following your request about a fair llama.cpp comparison of our CAT-Q with IQ1_Quant (specifically IQ1_S and IQ1_M), now we have made additional experimental results available, including hardware profiling, perplexity on WikiText2, and accuracy on five zero-shot commonsense reasoning tasks evaluated in our paper. For consistency, all additional experiments are conducted under the same llama.cpp deployment framework and hardware setup as those reported in Table B and C of our rebuttal, and the same performance benchmarking settings as those reported in Table 1 of our paper and Table A. Recall that we deployed our 1.58-bit Qwen3-4B and Llama2-7B models using llama.cpp with its ternary implementation (TQ2_0 for GPU and TQ1_0 for CPU). Your mentioned IQ1_S and IQ1_M are applicable to both GPU and CPU. Tables D and E below summarize the detailed results. We can observe that, for both Qwen3-4B and Llama2-7B: **1)** in terms of model accuracy, our CAT-Q outperforms IQ1_Quant (IQ1_S and IQ1_M) with significant margins on both WikiText2 and five commonsense reasoning tasks; **2)** in terms of decoding speed and memory footprint, our 1.58-bit models are mostly on par with those by IQ1_Quant (IQ1_S and IQ1_M), on both GPU and CPU. These results further demonstrate the merits of our CAT-Q.
> > >
> > > **Table D**: Model profiling with llama.cpp on an NVIDIA RTX 4090 GPU and an Intel Core i9-13900KF CPU, together with model performance on WikiText2 (perplexity) and five commonsense reasoning tasks (averaged accuracy).
> > > |Model|Quantization Method|Weight Memory on GPU\|CPU (GB)|Decoding Throughput on GPU\|CPU (tokens per sec)|WikiText2↓|Avg↑|
> > > |-|:-:|:-:|:-:|:-:|:-:|
> > > |Qwen3-4B|IQ1_S|**1.00\|1.00**|**322.23\|37.54**|70.90|39.31|
> > > |Qwen3-4B|IQ1_M|1.06\|1.06|312.87\|36.99|35.86|42.38|
> > > |Qwen3-4B|CAT-Q (ours)|1.17\|1.01|320.34\|37.35|**18.99**|**57.06**|
> > > |Llama2-7B|IQ1_S|**1.42\|1.42**|**307.33\|27.44**|24.33|40.18|
> > > |Llama2-7B|IQ1_M|1.54\|1.54|301.01\|25.10|19.35|44.32|
> > > |Llama2-7B|CAT-Q (ours)|1.90\|1.62|281.66\|26.30|**8.51**|**57.69**|
> > >
> > > **Table E**: Detailed results on five commonsense reasoning tasks (accuracy) and WikiText2 (perplexity).
> > > |Model|Quantization Method|WikiText2↓|PIQA↑|ARC-e↑|ARC-c↑|HellaSwag↑|Winogrande↑|Avg↑|
> > > |-|:-:|:-:|:-:|:-:|:-:|:-:|:-:|:-:|
> > > |Qwen3-4B|IQ1_S|70.90|55.22|34.72|23.55|31.43|51.62|39.31|
> > > |Qwen3-4B|IQ1_M|35.86|59.52|37.92|26.51|37.23|50.74|42.38|
> > > |Qwen3-4B|CAT-Q (ours)|**18.99**|**70.62**|**62.29**|**36.95**|**53.24**|**62.19**|**57.06**|
> > > |Llama2-7B|IQ1_S|24.33|59.40|33.60|23.36|38.75|49.80|40.18|
> > > |Llama2-7B|IQ1_M|19.35|62.40|38.26|25.55|43.00|52.40|44.32|
> > > |Llama2-7B|CAT-Q (ours)|**8.51**|**72.91**|**60.06**|**33.62**|**60.95**|**60.93**|**57.69**|
> > >
> > > Thanks again for your constructive comments, time and patience.
> > >
> > > Looking forward to your feedback!

---

### Official Review · Reviewer_fHJg · 2026-03-12

**Soundness:** 3
**Presentation:** 4
**Significance:** 3
**Originality:** 3
**Overall Recommendation:** 6
**Confidence:** 4

**Summary:**

This paper presents a Cost-effective Accurate Ternary Quantization (CAT-Q) method for ternary LLMs. They achieve a good trade-off between accuracy and Post-Quantization Training (PTQ) cost, while balancing hardware execution efficiency. They select W = aT as the final approximation, but a more complex W representation that includes u and a in the PTQ. They propose soft ternarization with a differential formulation to overcome the drawbacks of the hard ternarization problem. They also apply the cross-layer quantization to ternary LLMs. The evaluation results show that the CAT-Q achieves higher model accuracy on 10 models (1.7B to 235B) 5 datasets compared with many related works. CAT-Q only needs 1M or 2M training tokens for quantization, which is much fewer than other works.

This work will greatly advance the adoption of ternary LLMs because it can provide accurate models with very low PTQ cost. This method is also applicable to very large models like QWen3-235B-A22B. This paper makes a significant contribution to the field of LLM quantization. Therefore, I recommend a Strong Accept.

**Compliance With Llm Reviewing Policy:**

Affirmed.

**Final Justification:**

This is a good paper. So I recommend strong accept in the beginning and also in the end. Other reviewers with Weak Reject changed to Accept or Weak Accept due to their additional experiments and clarifications too.

**Key Questions For Authors:**

+ In Table 2, Qwen3 + CAT-Q models have 1 M or 2 M training tokens for quantization, and using more tokens yields higher accuracy. But other models use 100B or more tokens, which is unfair. What if you use 1B or even 100B tokens?
+ Table 3 has similar but different model parameter settings on BitNet and CAT-Q. How do you know the improvement comes from the CAT-Q not the larger model size?
+ Why Table 6 does not cover grama = 0.6 and 0.7?
+ Does the group size of 128 mean that every 128 values in a column of the weight matrix share the same scaling factor? Suppose the X is M*K and the weight is K*N, then each column has K/128 scaling factors? Or do 128 columns share the same scaling factor?

**Limitations:**

No. The authors have not discussed the limitations.

**Strengths And Weaknesses:**

Strength:
+ Good novelty. Though the learnable scaling factors are very common in TWNs, the differential quantization function to mitigate the hard ternarization is a new concept to me. They also apply the cross-layer quantization to ternary LLMs.
+ Very thorough evaluation. This paper covers all aspects of model sizes (1.7B to 235B), a series of related works, ablation studies on PTQ hyperparameters, etc.

Weakness:
- The code will be available... Not provided for review.

---

> ### Author Rebuttal · Authors · 2026-03-31
>
> Thank you for the constructive comments and the recognition of our work.
>
> **1. To your first Key Question** about the performance of CAT-Q using more tokens for calibration in Table 2.
>
> **Our responses**: In Table 2, we use Qwen3+CAT-Q models using 1M vs. 2M calibration tokens to illustrate the performance scaling behavior of our method w.r.t. the amount of calibration tokens. The existing SOTA 1.58-bit LLMs in Table 2 including BitNet V1/V2, TriLM and Tequila all adopt QAT, which are trained with e.g., 100B or 300B training tokens, incurring prohibitive training costs for most researchers. Yes, compared to them, it is unfair to our CAT-Q using only 1M calibration tokens by default, but it has shown our method's advantage in terms of quantization cost and accuracy. To further validate this behavior, in Table A below, we conduct an experiment using **8M** tokens (which costs ~16 hours on 8 A100-80GB GPUs), yielding 3.72% gain over 2M tokens. Larger gains could be expected by using 1B or 100B tokens, but we are unable to verify it due to limited computational resources.
>
> **Table A**: The performance scaling behavior of our method w.r.t. the amount of calibration tokens.
>
> |Model|#Tokens|PIQA↑|ARC-e↑|ARC-c↑|HellaSwag↑|Winogrande↑|Avg↑|
> |-|:-:|:-:|:-:|:-:|:-:|:-:|:-:|
> |Qwen3-4B|1M|69.80|61.53|36.95|53.98|60.69|56.59|
> |Qwen3-4B|2M|71.55|66.41|39.08|55.01|61.96|58.80|
> |Qwen3-4B|8M|**72.17**|**71.64**|**43.41**|**61.39**|**63.98**|**62.52**|
>
> **2. To your second Key Question** about the accuracy improvement of our 1.58-bit models over BitNetV2 models in Table 3.
>
> **Our responses**: Existing SOTA 1.58-bit LLMs including BitNetV2 compared in Table 3 all adopt QAT, which are trained from scratch. They are tailored for a specific LLM architecture family and small model sizes, and they do not release their FP16 models to public. This makes an apples-to-apples comparison with them is infeasible although our CAT-Q uses PTQ and demonstrates appealing scalability to diverse LLM architectures and sizes. In Table B below, we conduct extra experiments to compare CAT-Q and BitNet on the same-sized 7B models, showing that CAT-Q with 2M (1M) calibration tokens outperforms (is superior/competitive to) BitNetV1/V2 trained with 100B tokens.
>
> **Table B**: Comparison of CAT-Q and BitNet on the same-sized 7B models.
>
> |Model|#Bits|#Tokens|PIQA↑|ARC-e↑|ARC-c↑|HellaSwag↑|Winogrande↑|Avg↑|
> |-|:-:|:-:|:-:|:-:|:-:|:-:|:-:|:-:|
> |BitNetV1-7B|W1.58A16|100B|**74.37**|59.51|31.74|61.49|59.98|57.42|
> |Llama2-7B+CAT-Q|W1.58A16|1M|72.91|60.06|33.62|60.95|60.93|57.69|
> |Llama2-7B+CAT-Q|W1.58A16|2M|74.25|**63.07**|**36.29**|**63.12**|**63.75**|**60.10**|
> |BitNetV2-7B|W1.58A8|100B|**74.10**|58.54|32.94|61.08|61.48|57.63|
> |Llama2-7B+CAT-Q|W1.58A8|1M|72.45|59.78|33.15|60.12|60.23|57.15|
> |Llama2-7B+CAT-Q|W1.58A8|2M|73.96|**62.81**|**35.89**|**62.91**|**63.55**|**59.82**|
>
> **3. To your third Key Question** about why Table 6 does not cover gamma = 0.6 and 0.7?
>
> **Our responses**: This is because we were not aware of adopting a uniform sampling strategy to select values of $\gamma$ for the ablation in Table 6. In Table C below, we improve this ablation by including $\gamma=0.6|0.7$. We can see that the default setting $\gamma=0.8$ always gets the best results; $\gamma=0.7$ yields results similar to $\gamma=0.9$ and slightly better than $\gamma=0.9$.
>
> **Table C**: Ablation on varying choices of $\gamma$.
>
> |Model|Ratio $\gamma$|PIQA↑|ARC-e↑|ARC-c↑|HellaSwag↑|Winogrande↑|Avg↑|
> |-|:-:|:-:|:-:|:-:|:-:|:-:|:-:|
> |Qwen3-4B|0.5|68.15|60.95|34.81|51.33|58.98|54.84|
> |Qwen3-4B|0.6|69.04|61.24|35.75|51.77|61.80|55.92|
> |Qwen3-4B|0.7|69.04|61.81|36.25|52.37|61.90|56.27|
> |Qwen3-4B (default setting)|0.8|**70.62**|**62.29**|**36.95**|**53.24**|**62.19**|**57.06**|
> |Qwen3-4B|0.9|69.75|61.80|35.36|52.47|61.46|56.17|
> |Qwen3-4B|1.0|68.77|59.55|34.98|51.55|60.27|55.02|
> |Llama2-7B|0.5|69.12|57.18|30.12|59.73|59.91|55.21|
> |Llama2-7B|0.6|70.56|58.81|31.10|60.13|60.18|56.16|
> |Llama2-7B|0.7|72.13|59.18|32.89|**61.71**|60.91|57.36|
> |Llama2-7B (default setting)|0.8|**72.91**|**60.06**|**33.62**|60.95|**60.93**|**57.69**|
> |Llama2-7B|0.9|71.56|59.24|33.13|59.90|61.48|57.06|
> |Llama2-7B|1.0|70.67|58.80|31.38|60.02|59.04|55.98|
>
> **4. To your last Key Questions** about the group-size (GS).
>
> **Our responses**: We follow GS=128 used in recent 1.58-bit QAT works TriLM and Tequila. Yes, the GS of 128 means every 128 values in a column of the weight matrix share the same scaling factor, and for your mentioned case, each column has $K/128$ scaling factors. Table 11 of our paper provides an ablation to study varying choices of GS, showing our method also achieves good performance with channel-wise ternarization.
>
> **5**. As promised in our paper, we are currently preparing the code for public release to advance 1.58-bit LLM quantization research.
>
> **6.** We also add a discussion on limitations (see our last set of responses to Reviewer eGPr) and more experiments (see our responses to other reviewers).

---

> > ### Author Rebuttal · Reviewer_fHJg · 2026-04-02
> >
> > Thank you for the clear responses. This paper definitely worth accepted by ICML. I suggest you to release all the models you have fine-tuned (e.g., the Llama2-7B+CAT-Q, QWen3-4B+CAT-Q, and each in best fine-tuned case) together with the fine-tuning code.

---

> > > ### Author Response · Authors · 2026-04-07
> > >
> > > We sincerely appreciate your strong recognition of our work as well as our rebuttal. We are pleased to support the 1.58-bit LLM research community and will be releasing our code and all trained models shortly.
> > >
> > > Thanks again for your constructive comments, time and patience.
> > >
> > > The authors.

---

### Official Review · Reviewer_TC1v · 2026-03-13

**Soundness:** 3
**Presentation:** 3
**Significance:** 3
**Originality:** 3
**Overall Recommendation:** 5
**Confidence:** 4

**Summary:**

The paper introduces CAT-Q, a post-training quantization (PTQ) method designed to compress Large Language Models (LLMs) into 1.58-bit ternary representations. Unlike existing state-of-the-art ternary methods that rely on expensive, data-intensive QAT, CAT-Q employs a PTQ approach. It utilizes two main components: Learnable Modulation, which adaptively transforms weight distributions using learnable factors to mitigate distributional misalignment, and Softened Ternarization, a two-stage process that uses a novel smooth transition function to guide the weights from full precision toward a ternary state during calibration. Experimental results across a diverse set of models (up to 235B parameters) suggest that CAT-Q achieves competitive performance with significantly fewer tokens and lower computational overhead.

**Compliance With Llm Reviewing Policy:**

Affirmed.

**Final Justification:**

The authors provided additional experiments and discussions, which resolved most of my concerns. Therefore, I decided to raise the rating to "5: Accept".

**Key Questions For Authors:**

Please refer to the weaknesses.

**Limitations:**

Yes

**Strengths And Weaknesses:**

Strengths:
- By pivoting from QAT to a PTQ-based framework, the authors achieve massive savings in compute resources and training time, making ternary quantization more accessible.
- The empirical study is quite broad, covering 10 different models, including dense and MoE architectures, and demonstrating scalability up to 235B parameters.

Weaknesses

My concerns are primarily focused on the evaluation and methodology:

- The method relies on specific hyper-parameters (e.g., \gamma=0.8, s_0=30). TThe authors designate these as default settings but provide no systematic analysis of their sensitivity across various model scales and architectures. Are these parameters universal across diverse model families and sizes, or do they require per-model tuning? The current lack of cross-model hyper-parameter ablation raises concerns regarding the method's practical robustness and generalizability.
- The paper benchmarks against BitNet 1.58-bit models, which are trained from scratch. Comparing a PTQ-finetuned model (which leverages the high-quality representations of pre-trained Qwen3 weights) against a model trained from scratch is fundamentally unbalanced. The authors should explicitly clarify this disparity in their methodology to avoid potentially misleading claims regarding "competitive performance."
- The adoption of the sliding-layer optimization framework from SliderQuant might be a significant component of the final performance. However, the paper fails to isolate the gain attributed to this framework versus the proposed LM and ST modules. To demonstrate the true value of the authors' specific contributions, an ablation study isolating the effects of the sliding-layer reconstruction is essential. Without it, the performance boost may be unfairly credited to the authors' novel components.

---

> ### Author Rebuttal · Authors · 2026-03-31
>
> Thank you for the constructive comments and the recognition of our work.
>
> **1. To your first Concern** about the lack of cross-model ablations of hyper-parameters $\gamma$ and $s_0$.
>
> **Our responses**: Indeed, for our method, $\gamma$ and $s_0$ are two critical hyper-parameters. $\gamma$ controls the ratios of calibration epochs allocated to the two stages (differentiable ternarization and hard ternarization) of our method, and $s_0$ controls the final sharpness of our proposed smooth transition function $f(\cdot)$ for the differentiable ternarization stage. In our main experiments, your mentioned default settings ($\gamma=0.8, s_0=30$) are universal across all LLM families and sizes, without per-model tuning. **We actually provided two cross-model ablations** comparing varying choices of $\gamma$ and $s_0$ with Qwen3-4B and Llama2-7B in Table 6 (Main paper) and Table B & Figure A (Appendix), respectively. In **Table A and B below**, we further improve these two ablations (* denotes newly added experiments), which validate the practical robustness and generalizability of our method (gets stable performance when $\gamma=[0.7,0.9]$ and $s_0=[15,100]$).
>
> **Table A**: Ablation on varying choices of $\gamma$.
>
> |Model|Ratio $\gamma$|PIQA↑|ARC-e↑|ARC-c↑|HellaSwag↑|Winogrande↑|Avg↑|
> |-|:-:|:-:|:-:|:-:|:-:|:-:|:-:|
> |Qwen3-4B|0.5|68.15|60.95|34.81|51.33|58.98|54.84|
> |Qwen3-4B (*)|0.6|69.04|61.24|35.75|51.77|61.80|55.92|
> |Qwen3-4B (*)|0.7|69.04|61.81|36.25|52.37|61.90|56.27|
> |Qwen3-4B (default setting)|0.8|**70.62**|**62.29**|**36.95**|**53.24**|**62.19**|**57.06**|
> |Qwen3-4B|0.9|69.75|61.80|35.36|52.47|61.46|56.17|
> |Qwen3-4B|1.0|68.77|59.55|34.98|51.55|60.27|55.02|
> |Llama2-7B|0.5|69.12|57.18|30.12|59.73|59.91|55.21|
> |Llama2-7B (*)|0.6|70.56|58.81|31.10|60.13|60.18|56.16|
> |Llama2-7B (*)|0.7|72.13|59.18|32.89|**61.71**|60.91|57.36|
> |Llama2-7B (default setting)|0.8|**72.91**|**60.06**|**33.62**|60.95|**60.93**|**57.69**|
> |Llama2-7B|0.9|71.56|59.24|33.13|59.90|61.48|57.06|
> |Llama2-7B|1.0|70.67|58.80|31.38|60.02|59.04|55.98|
>
> **Table B**: Ablation on varying choices of $s_0$.
>
> |Model|$s_0$|PIQA↑|ARC-e↑|ARC-c↑|HellaSwag↑|Winogrande↑|Avg↑|
> |-|:-:|:-:|:-:|:-:|:-:|:-:|:-:|
> |Qwen3-4B|10|69.42|59.68|35.32|51.34|59.04|54.96|
> |Qwen3-4B(*)|15|69.61|60.72|35.79|52.53|60.83|55.90|
> |Qwen3-4B (default setting)|30|**70.62**|**62.29**|**36.95**|53.24|**62.19**|**57.06**|
> |Qwen3-4B|100|69.26|61.24|35.75|**53.33**|61.48|56.21|
> |Qwen3-4B(*)|200|68.63|58.07|33.19|48.27|58.96|53.42|
> |Qwen3-4B|1000|66.59|53.28|31.31|45.48|57.46|50.82|
> |Llama2-7B|10|70.46|54.55|31.48|57.57|56.51|54.11|
> |Llama2-7B(*)|15|71.28|56.88|32.41|59.12|58.46|55.63|
> |Llama2-7B (default setting)|30|**72.91**|**60.06**|**33.62**|**60.95**|**60.93**|**57.69**|
> |Llama2-7B|100|72.42|59.46|33.14|59.94|59.51|56.89|
> |Llama2-7B(*)|200|70.72|56.33|31.65|57.43|57.41|54.71|
> |Llama2-7B|1000|66.65|45.88|27.13|50.11|56.12|49.18|
>
> **2. To your second Concern about** explicitly clarifying the disparity in the methodology of existing SOTA 1.58-bit models and ours in performance benchmark.
>
> **Our responses**: We fully agree with your insightful comments. In 1.58-bit LLM quantization research, existing SOTA models including BitNet V1/V2, TriLM and Tequila that we compared all adopt QAT, which are trained from scratch. They are tailored for a specific LLM architecture family and small model sizes, and they do not release their FP16 models to public. This makes an apples-to-apples comparison with them is infeasible although our method adopts PTQ and demonstrates appealing scalability to diverse LLM architectures and sizes. Following your suggestion, we will explicitly clarify this disparity in the methodology of existing SOTA 1.58-bit models and ours when stating "competitive performance".
>
> **3. To your third Concern** about missing an ablation study isolating the effects of the sliding-layer reconstruction.
>
> **Our responses**: The sliding-layer reconstruction (SLR) from SliderQuant (ICLR 2026) is used as the base framework of our CAT-Q, as we clarified in the Method section 2.4. **We actually provided a cross-model ablation study** with Qwen3-4B and Llama2-7B in **Table 5** (Main paper), isolating the effects of the SLR framework and comparing the gain attributed to it versus our LM (Learnable Modulation) and ST (Softened Ternarization) modules. Concretely, in Table 5, for Qwen3-4B/Llama2-7B, **the results in the first row are for the SLR framework**, we can see: (1) applying LM over SLR improves the averaged accuracy from 40.16%/45.37% to 53.19%/55.97%, achieving 13.03%/10.60% gain; (2) by further applying ST over LM+SLR, the performance is boosted to 57.06%/57.69%. These results validate the importance of our core modules LM and ST. We are sorry for missing the name of SLR in Table 5. We will remove this confusion in paper revision.
>
> **4.** As for other experiments and discussions that we have made during the rebuttal phase, you are referred to our responses to the other three reviewers.

---

> > ### Author Rebuttal · Reviewer_TC1v · 2026-04-07
> >
> > Thanks for the response. The authors provided detailed experiments which fully addressed my concerns. I will raise my rating.

---

> > > ### Author Response · Authors · 2026-04-08
> > >
> > > We sincerely appreciate your recognition of our work and our rebuttal.
> > >
> > > Thank you again for your constructive review, which is highly valuable in improving the quality of our work, as well as for your time and patience.
> > >
> > > The authors.

---

### Decision · Program_Chairs · 2026-04-30

**Decision:**

Accept (spotlight)

**Comment:**

This paper proposes CAT-Q, a post-training quantization method for compressing LLMs into 1.58-bit ternary representations. It introduces Learnable Modulation and Softened Ternarization to address distribution misalignment and convergence instability. It achieves competitive performance with a significantly lower computational cost.


The main strengths of this paper are:
- High cost-efficiency and practicality, making ternary quantization more accessible (Reviewer TC1v, eGPr, LXB9).
- Comprehensive empirical validation across 10 diverse models (including architectures and model sizes) on 5 datasets (Reviewer TC1v, fHJg, LXB9).
- High novelty in using a differential quantization function to mitigate hard ternarization (Reviewer fHJg)

The main weaknesses are:
- Lack of discussions/ablations regarding hyper-parameters (Reviewer TC1v) and isolating the effects of the sliding-layer reconstruction (Reviewer TC1v, LXB9)
- Unclear claims due to pretraining/training from scratch (Reviewer TC1v)
- Unclear improvement related to token/model size (Reviewer fHJg),
- Lack of more evaluation benchmarks (Reviewer LXB9) and evaluation on target hardware/real-world deployment (Reviewer eGPr)


Overall, this paper received 4 positive scores (1 Strong Accept, 2 Accept, 1 Weak Accept). All the concerns are addressed during the rebuttal with solid experiments. The AC agrees with its novelty, its superior performance, and tends to accept. The authors are suggested to provide an improvement in final version based on the reviews.